# Eating for a Sustainable Planet: Personalized Sustainable Diet Recommendation via Constraint-Aware Decision-Making Modeling

**Ying Jin** [1 2 3]  **Weiqing Min** [2 3]  **Mingyu Huang** [2 3]  **Shuqiang Jiang** [2 3]

## Abstract

A sustainable diet represents a multi-dimensional synergy among four essential pillars: nutrition adequacy, economic affordability, cultural acceptability, and environmental respect. Despite the prevalence of population-level sustainability modeling, practical implementation relies on effective individual-level adoption. This transition is often hindered by inter-individual heterogeneity, posing a formidable challenge in aligning sustainable diet requirements with individual preferences. To address this issue, we propose a personalized sustainable diet recommendation model based on a constraint-aware decision-making mechanism, where sustainability is incorporated through learnable constraints rather than modeled as user preferences. To systematically evaluate the proposed approach, we construct a sustainable diet dataset named SusDiet with about 150k recipes, characterized by broad coverage of sustainability indicators. Experimental results on this dataset show that our method promotes more sustainable choices without compromising individual preference. This work establishes a framework for aligning individual dietary choices with planetary health, offering quantitative evidence to guide future sustainable diet interventions and policy-making for sustainable development.

## 1. Introduction

Sustainable development stands as a defining global challenge of the 21st century (Sachs, 2012). In response to accelerating climate change, biodiversity loss, and deepening social inequality, the United Nations' Sustainable Development Goals (SDGs) provide a shared framework for global progress (Lee et al., 2016; Colglazier, 2015; Hák et al., 2016). Among the SDGs, climate action, responsible consumption and production, zero hunger, good health and well-being, clean water use, and life on land are all closely related to dietary patterns (Rockström et al., 2009b;a; Steffen et al., 2015). Consequently, dietary change has been recognized as a key leverage point to address global sustainability challenges and advance multiple SDGs simultaneously (Springmann et al., 2018; Willett et al., 2019).

Within this sustainability agenda, sustainable diets are defined as dietary patterns that promote individual health and well-being while simultaneously ensuring low environmental impact, economic affordability, safety, and cultural acceptability (Organization et al., 2021; Gibney, 2025). However, translating this consensus into feasible, widely adopted practices remains an open challenge (Biesbroek et al., 2023). To address this, extensive research has sought to define the multifaceted requirements of sustainable dietary patterns (Springmann et al., 2018; Bajželj et al., 2021; Ye et al., 2024; Merrigan et al., 2015). Current studies often utilize mathematical optimization or scenario analysis to meet nutritional, environmental, economic, and cultural objectives within idealized constraints (Springmann et al., 2018; Gazan et al., 2018; Wilson et al., 2019). While such studies provide foundational guidance for dietary transitions at national or global scales (Heerschop et al., 2024), they primarily treat sustainability as an explicit objective or hard constraint at the population level, while neglecting inter-individual heterogeneity. This focus limits the application of global recommendations to real-world sustainable practices, highlighting the need to model sustainable dietary patterns at the individual level (de Costa et al., 2025).

Existing dietary patterns are shaped by long-standing cultural norms, economic constraints, and deeply ingrained eating habits, rendering the transition to sustainable diets a formidable challenge for the individual. Even when minimizing deviations from current dietary patterns, achieving nutritionally adequate and environmentally sustainable diets often still requires major structural changes in consumption,

---

[1]School of Advanced Interdisciplinary Sciences, University of Chinese Academy of Sciences, Beijing, China [2]State Key Laboratory of AI Safety, Institute of Computing Technology, Beijing, China [3]University of Chinese Academy of Sciences, Beijing, China. Correspondence to: Weiqing Min <minweiqing@ict.ac.cn>.

*Proceedings of the 43rd International Conference on Machine Learning*, Seoul, South Korea. PMLR 306, 2026. Copyright 2026 by the author(s).

particularly regarding staple foods, meat, and dairy products (Chaudhary & Krishna, 2019; Wilson et al., 2019). These changes often exceed the threshold of individual acceptability. As a result, consumers often exhibit low willingness to change their dietary behavior, even when aware of the necessity (Elliott et al., 2024; Gibney, 2025; Biesbroek et al., 2023), making it difficult to map conceptual findings onto the everyday decision-making of real individuals.

To bridge this gap, we need modeling approaches focused on individual decision-making. However, current personalized diet recommendation systems, largely adapted from other domains, focus primarily on learning and optimizing for explicit user preferences or satisfaction (Qiao et al., 2025; Zhang et al., 2023; Yang et al., 2017). This paradigm implicitly assumes that food choices are driven by consciously articulated tastes and interests, and consequently treats sustainability also as an explicit preference dimension. In reality, individuals rarely make dietary decisions with explicit preference or awareness regarding sustainability (Fernqvist et al., 2024; Gonzalez et al., 2025; Nguyen et al., 2025). Therefore, treating sustainability merely as a preference to learn is conceptually misaligned with human behavior. Consequently, developing a decision-making framework that acknowledges individual differences while adhering to sustainability constraints rather than simply optimizing for user preferences remains a major challenge.

To address this challenge, we propose a novel framework that reformulates personalized sustainable diet recommendation as a constraint-aware decision-making process. Unlike approaches that model sustainability as explicit user preference, our method formulates sustainability in terms of learnable constraint thresholds and jointly integrates them with preference learning within a unified decision-making framework. In our formulation, cultural acceptability is implicitly reflected through user preference representations learned from historical dietary interactions, while other sustainability dimensions, including nutrition, environmental impact, economic cost, and animal welfare, are modeled through explicit measurable indicators. The framework consists of three technically distinct components: (1) A **Mixture-of-Experts (MoE) Transformer** to encode multi-dimensional sustainability signals from structured ingredient compositions and quantitative attributes. It can produce expert-specific embeddings that facilitate the learning of constraint encodings across heterogeneous sustainability dimensions. These representations are supervised through an uncertainty-weighted multi-task learning objective to ensure balanced modeling of diverse sustainability indicators. (2) A **multi-interest attention mechanism** to capture heterogeneous taste preferences from historical interactions, enabling the model to represent multiple latent preference profiles for each user. (3) A **personalized constraint mechanism** that learns user-specific sustainability

constraint thresholds and trade-off weights. These constraints interact with sustainability signals to dynamically penalize recipes that exceed individual sustainability boundaries in the final ranking score. The overall framework is optimized with a unified objective that jointly balances preference learning and sustainability constraint enforcement, enabling effective trade-offs between personalization accuracy and multi-dimensional sustainability considerations.

In addition, we construct **SusDiet**, a specialized sustainability dataset designed to support large-scale modeling of sustainability-aware dietary decision-making. SusDiet contains 149,036 recipes annotated with a rich collection of sustainability indicators across multiple dimensions, including nutritional adequacy, environmental impact, economic cost, and animal welfare, providing a holistic characterization of recipe level sustainability. These recipe annotations are further integrated with real-world user recipe interaction data, resulting in a dataset comprising 179,869 users and 744,138 interaction records. Extensive experiments conducted on SusDiet demonstrate the effectiveness of the proposed approach.

Our specific contributions are as follows:

- We propose a personalized sustainable diet recommendation framework, formulated as a constraint-aware decision-making task. By integrating a MoE-based sustainability model, a multi-interest attention mechanism for preferences learning and a personalized constraint mechanism for user-specific sustainability trade-offs, the framework effectively balances preference learning with sustainability requirements to deliver tailored, eco-friendly recommendations.

- We construct a large-scale real-world dataset with approximately 150k recipes annotated by comprehensive sustainability indicators across nutritional adequacy, environmental impact, economic cost, and animal welfare, aligned with user-recipe interactions.

- Experiments demonstrate that learning user-specific sustainability constraints yields substantially more sustainable recommendations while maintaining competitive recommendation accuracy, offering quantitative evidence for both modeling effectiveness and policy relevance.

## 2. Related Work

Research on sustainable diets has emerged as a critical domain addressing the nexus of human health and planetary boundaries. Existing research on sustainable diets can be broadly categorized into macro-level optimization aimed at policy formulation and micro-level modeling focused on individual behavioral interventions (Biesbroek et al., 2023).

**Macro-level Modeling and Optimization** A dominant line of work operates at the population or national level, utilizing large-scale modeling to develop dietary patterns that meet nutritional needs while minimizing environmental impacts and considering cost or cultural constraints. For example, the health and environmental benefits of shifting toward plant-based diets have been quantified across 150 countries (Springmann et al., 2018), while a nonlinear optimization framework was proposed to design country-specific diets that minimize deviations from current consumption under planetary-boundary constraints (Chaudhary & Krishna, 2019). Similarly, region-specific reference diets have been derived through extensive data analysis (Ye et al., 2024) and machine learning has been widely used to navigate trade-offs among nutrition, environmental impact, cost, and acceptability (Wilson et al., 2019; Gazan et al., 2018). While these policy-centric and optimization-based approaches provide crucial evidence for the feasibility of sustainable food systems and inform global strategies, they are inherently normative. The output of these approaches is a macro-level optimal pattern rather than a model of how sustainable choices emerge from heterogeneous individuals' decision processes.

**Individual-level Recommendations** Complementing macro-level prescriptions, individual-level modeling offers a path toward scalable, personalized interventions. By learning preferences at scale, recommendation systems (RS) can deliver tailored dietary guidance aligned with sustainability goals (He et al., 2024). Recent efforts include datasets and methods aimed at sustainability-oriented recommendations (Zhang et al., 2024a), but these studies focus primarily on nutritional and environmental dimensions. Many food RS still prioritize recommendation accuracy; for instance, an RS model was proposed to improve performance without explicitly promoting sustainability (Zhang et al., 2024b). More behavior-aware approaches attempt to integrate greener choices into preference learning. For example, Jing et al. (2025) proposed a recommendation framework to capture user preferences and attitudes toward greener options. Yet, such methods often presume that users hold explicit sustainability attitudes, which is not always true in reality. In contrast, our constraint-aware decision-making mechanism incorporates sustainability dimensions without requiring them to be explicitly represented in user preferences, thus addressing this limitation.

## 3. Task Formulation

We study the problem of personalized sustainable diet recommendation. The goal is to recommend recipes that satisfy a user's taste preferences while adhering to their unique boundaries across various sustainability metrics. Beyond the conventional recommendation objective of learning user

preferences, this task formulation requires the modeling of user-specific constraints for each sustainability dimension.

Formally, we assume there are $N$ sustainability dimensions. Each recipe $r$ is associated with an $N$-dimensional sustainability indicator vector $\boldsymbol{s}_r = [s_r^1, \ldots, s_r^N]$, where each dimension corresponds to a specific sustainability criterion.

For each user $u$, we define a user-specific constraint vector $\boldsymbol{\lambda}_u = [\lambda_u^1, \ldots, \lambda_u^N]$, where $\lambda_u^n$ denotes the bound on the $n$-th sustainability indicator that the user is willing to tolerate. Without loss of generality, all sustainability indicators are normalized and formulated in a unified minimization form, such that lower values indicate more sustainable outcomes.

Given an estimated preference score $\hat{y}_{u,r}$ for each user–recipe pair $(u, r)$, where larger values indicate stronger preference, the personalized recommendation problem can be formulated as the following constrained optimization:

$$\max_r \quad \hat{y}_{u,r}$$
$$\text{s.t.} \quad s_r^n \le \lambda_u^n, \quad n = 1, \ldots, N. \tag{1}$$

To handle these constraints, we introduce a user-specific Lagrange multiplier vector $\boldsymbol{\alpha}_u = [\alpha_u^1, \ldots, \alpha_u^N]$, and define the following function:

$$F(u, r) = \hat{y}_{u,r} - \boldsymbol{\alpha}_u^\top (\boldsymbol{s}_r - \boldsymbol{\lambda}_u). \tag{2}$$

Our objective is to maximize $F(u, r)$. Accordingly, the task is defined as learning user-specific sustainability constraints $\boldsymbol{\lambda}_u$ and trade-off weights $\boldsymbol{\alpha}_u$, and recommending the top-$K$ recipes that maximize $F(u, r)$ for each user.

## 4. Methodology

Based on the proposed formulation, we design a unified framework for personalized sustainable diet recommendation, as illustrated in Figure 1.

The framework consists of three synergistic components: (i) a recipe sustainability representation that encodes each recipe into a latent embedding $\mathbf{z}_r \in \mathbb{R}^D$ using a Mixture-of-Experts (MoE) architecture to capture heterogeneous and potentially competing sustainability indicators such as nutritional adequacy, environmental impact, economic cost, and animal welfare; (ii) a preference learning that estimates a preference score $\hat{y}_{u,r}$ for each user–recipe pair $(u, r)$ solely from historical interactions, thereby modeling intrinsic user interests independently of sustainability considerations; and (iii) a personalized constraint-aware trade-off that integrates the learned recipe representations and preference scores to infer user-specific decision parameters, namely a constraint encoding $\boldsymbol{\lambda}_u$ and a trade-off encoding $\boldsymbol{\alpha}_u$, thereby enabling balance between individual preferences and personalized sustainability boundaries.

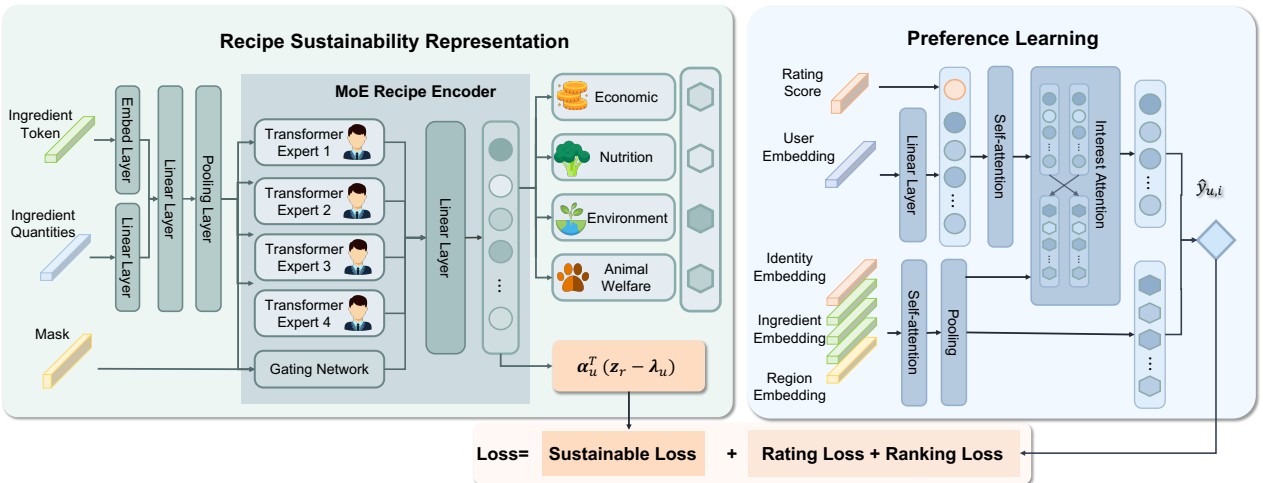

*Figure 1.* **The proposed framework for personalized sustainable diet recommendation.** The architecture including three components: (1) The **Recipe Sustainability Representation** (left) utilizes a MoE encoder to transform ingredient tokens and quantities into a multi-dimensional sustainability embedding $\mathbf{z}_r$. (2) The **Preference Learning** (right) leverages self-attention and interest attention to output the preference score $\hat{y}_{u,i}$ from user-recipe interactions. (3) The **Personalized Constraint-aware Trade-off** (bottom) synthesizes these outputs to instantiate the decision function via user-specific parameters $\boldsymbol{\lambda}_u$ and $\boldsymbol{\alpha}_u$, as shown in the term $\boldsymbol{\alpha}_u^\top(\mathbf{z}_r - \boldsymbol{\lambda}_u)$. The entire system is optimized using a joint loss including sustainability, rating, and ranking objectives.

## 4.1. Recipe Sustainability Representation

This module learns sustainability-aware recipe representations by jointly modeling ingredient compositions and multiple sustainability attributes. It produces a structured embedding $\mathbf{z}_r$ that captures both culinary semantics and sustainability profiles.

**Ingredient-Quantity Fusion.** For a recipe $r$, we represent its composition as a sequence of $L$ ingredient tokens $\mathbf{t}_r \in \mathbb{N}^L$ and corresponding quantities $\mathbf{q}_r \in \mathbb{R}^L$. Each ingredient $t_{r,j}$ is mapped to an embedding $\mathbf{e}_{r,j} \in \mathbb{R}^D$, while its quantity $q_{r,j}$ is projected into a continuous space $\mathbf{u}_{r,j} = W_q q_{r,j} + b_q \in \mathbb{R}^{d_q}$. We fuse these to form the input sequence $\mathbf{X}_r = [\mathbf{x}_{r,1}, \ldots, \mathbf{x}_{r,L}]$, where:

$$\mathbf{x}_{r,j} = W_{\text{in}}[\mathbf{e}_{r,j}; \mathbf{u}_{r,j}] + b_{\text{in}} \in \mathbb{R}^D. \tag{3}$$

**Mixture-of-Experts Recipe Encoder.** To capture diverse ingredient patterns, we employ a Mixture-of-Experts (MoE) encoder with $E = 4$ Transformer-based experts. Each expert $e$ independently processes $\mathbf{X}_r$ and produces an expert-specific representation $\mathbf{p}_r^{(e)}$ via masked mean pooling. A gating network computes selection weights $\mathbf{g}_r = \text{softmax}(\text{MLP}(\bar{\mathbf{x}}_r))$ based on the sequence average $\bar{\mathbf{x}}_r$. The final sustainability-aware embedding $\mathbf{z}_r$ is:

$$\mathbf{z}_r = \text{LayerNorm}\left(W_g \sum_{e=1}^{E} g_r[e]\mathbf{p}_r^{(e)} + b_g\right). \tag{4}$$

**Multi-task Sustainability Prediction.** Given $\mathbf{z}_r$, we jointly predict $K$ sustainability indicators using task-specific heads: $\hat{s}_r^{(k)} = \text{MLP}_k(\mathbf{z}_r)$. To stabilize training across heterogeneous target scales, we apply a signed logarithmic transformation and z-score normalization to the raw targets $s_r^{(k)}$. We optimize the module using an uncertainty-weighted multi-task loss $\mathcal{L}_{\text{task}}$:

$$\mathcal{L}_{\text{task}} = \sum_{k=1}^{K}\left(\exp(-v_k) \cdot \text{MSE}(\hat{s}_r^{(k)}, s_r^{(k)}) + v_k\right), \tag{5}$$

where $v_k$ is a learnable parameter that automatically balances the gradient contribution of each task $k$ based on its respective noise level.

## 4.2. Preference Learning Module via Multi-interest

To capture heterogeneous and context-dependent user tastes, we design a multi-interest mechanism to learn preference-oriented representations from the historical interaction set $\mathcal{H}_u = \{(r_i, y_{u,r_i})\}_{i=1}^{N_u}$, where $y_{u,r_i}$ denotes the observed rating for recipe $r_i$ and $N_u$ is the interaction count.

**User and Recipe Encoding.** Each historical recipe $r_i$ is first mapped into a latent semantic space through a learnable embedding table $\mathbf{E}_u[r_i]$. To preserve chronological dependencies, positional embeddings are incorporated: $\mathbf{h}_i = \mathbf{E}_u[r_i] + \mathbf{p}_i$, where $\mathbf{p}_i$ denotes the embedding of the $i$-th position. We then apply MLP and self-attention over

the historical sequence to capture dependencies among interacted recipes and obtain refined representations $\{\tilde{\mathbf{h}}_i\}_{i=1}^{N_u}$.

For a target recipe $r$, we construct representations from recipe identity, ingredient composition, and region embeddings: $\mathbf{v}_{r,id} = \mathbf{E}_{item}[r]$, $\mathbf{v}_{r,ing} = \mathbf{E}_{ing}[r]$ and $\mathbf{v}_{r,reg} = \mathbf{E}_{reg}[r]$. Self-attention and pooling are further applied to obtain refined recipe representations, which are concatenated and projected into the final recipe embedding $\mathbf{e}_r$.

**Multi-interest Attention.** To capture diverse user interests from different recipe views, we apply multi-interest attention separately on the identity, ingredient, and regional representations. Taking $\tilde{\mathbf{v}}_{r,id}$ as the query, the attention weight for each historical interaction is computed by:

$$\beta_{i,r}^{id} = \frac{\exp\left(\phi_Q^{id}(\tilde{\mathbf{v}}_{r,id})^\top \phi_K^{id}(\tilde{\mathbf{h}}_i)/\sqrt{d}\right)}{\sum_{j=1}^{N_u} \exp\left(\phi_Q^{id}(\tilde{\mathbf{v}}_{r,id})^\top \phi_K^{id}(\tilde{\mathbf{h}}_j)/\sqrt{d}\right)}, \quad (6)$$

where $\phi(\cdot)$ represents linear projections. The interest-aware user representation is then aggregated as $\mathbf{e}_{u,id} = \sum_{i=1}^{N_u} \beta_{i,r}^{id} \phi_V^{id}(\tilde{\mathbf{h}}_i)$. Similarly, $\tilde{\mathbf{v}}_{r,ing}$ and $\tilde{\mathbf{v}}_{r,reg}$ are processed through the same attention mechanism to obtain the representation $\mathbf{e}_{u,ing}$ and $\mathbf{e}_{u,reg}$, respectively. The final user representation is computed by $\mathbf{e}_u = \text{MLP}\left([\mathbf{e}_{u,id}; \mathbf{e}_{u,ing}; \mathbf{e}_{u,reg}]\right)$.

Finally, the predicted preference score is defined by the inner product:

$$\hat{y}_{u,r} = \langle \mathbf{e}_u, \mathbf{e}_r \rangle. \quad (7)$$

### 4.3. Constraint-aware Trade-off

To enable constraint-aware trade-off during optimization, we first construct a user-specific sustainability penalty based on the learned sustainability representation:

$$p_{u,r} = \boldsymbol{\alpha}_u^\top (\mathbf{z}_r - \boldsymbol{\lambda}_u), \quad (8)$$

where $\alpha$ and $\lambda$ is learnable parameters. This term corresponds to the Lagrangian penalty in the task formulation, we incorporate it explicitly into the training objective to enforce sustainability constraints.

**Sustainability-aware Penalty Loss.** The sustainability loss is defined from the above penalty term:

$$\mathcal{L}_{sus} = \mathbb{E}_{(u,r)} \, p_{u,r}. \quad (9)$$

Minimizing this term encourages the model to reduce violations of user-specific sustainability constraints.

**Preference Learning Objective.** Preference learning is optimized by jointly modeling rating accuracy and ranking consistency. Specifically, we minimize a rating objective

that fits normalized explicit feedback,

$$\mathcal{L}_{rating} = \mathbb{E}_{(u,r)} \left(\hat{y}_{u,r} - y'_{u,r}\right)^2, \quad (10)$$

together with a ranking objective that enforces correct relative ordering of candidate recipes,

$$\mathcal{L}_{rank} = \mathbb{E}_{(u,r^+,r^-)} \left[-\log \sigma\left(\hat{y}_{u,r^+} - \hat{y}_{u,r^-}\right)\right], \quad (11)$$

where $(r^+, r^-)$ denotes a positive–negative recipe pair sampled from user interaction history.

**Overall Objective.** In addition to the above objectives, we apply $\ell_2$ regularization to constraint-related parameters to ensure stable optimization. The final training objective is defined as:

$$\mathcal{L} = \mathcal{L}_{rating} + \beta_1 \mathcal{L}_{rank} + \beta_2 \mathcal{L}_{sus} + \beta_3 \|\Theta\|_2^2, \quad (12)$$

where $\beta_1, \beta_2, \beta_3 \in [0,1]$ are hyperparameters, and $\|\Theta\|_2^2$ denotes $\ell_2$ regularization.

## 5. Construction of Dataset

To support the development and evaluation of sustainable diet recommendation, we constructed a comprehensive dataset SusDiet that aligns recipes with multi-dimensional sustainability indicators. These indicators are derived from authoritative sources covering nutritional adequacy, environmental impact, economic cost, and animal welfare. By further aligning these recipes with large-scale user interaction data, this dataset enables the study of personalized decision-making under sustainability considerations.

### 5.1. Data Collection and Processing

We construct a recipe dataset by integrating recipes from multiple public sources, including recipeDB (Batra et al., 2020), Yummly-28k (Min et al., 2016), Yummly-66k (Min et al., 2017), WorldCuisines (Winata et al., 2025) and World Wide Recipe (Magomere et al., 2025). Each recipe is characterized by a standardized ingredient list and a country label. Specifically, we parse the recipe text by segmenting it into individual ingredient phrases. Each phrase is structured into a standardized triplet consisting of quantity, unit, and ingredient name. Ingredient information is unified through LLM-assisted cleaning and processing. Country labels are standardized according to the United Nations M49 classification[1]. Based on this, these recipes are associated with authoritative sustainability indicators obtained from external databases covering multiple dimensions, including nutritional adequacy, environmental impacts (Clark et al.,

---

[1] https://unstats.un.org/unsd/methodology/m49/

*Table 1.* Summary statistics of the SusDiet

| Category | Count |
|---|---|
| Recipe | 149,036 |
| Ingredients | 7,340 |
| Units | 64 |
| Ingredient Phrases | 1,527,618 |
| Countries | 131 |
| User | 179,869 |
| Interaction | 744,138 |

2022), economic cost [2], and animal welfare[3].

## 5.2. LLM-Based Ingredient Structuring and Indicator Computation

To transform unstructured ingredient descriptions into a unified structured representation and to enable alignment with food product categories defined in sustainability indicator sources, we adopt an LLM-based processing method inspired by (Zhang et al., 2024a). Specifically, we employ the large language model GPT to perform component phrase analysis, quantity estimation, and sustainability indicator mapping. Ingredient phrases are decomposed into structured components of quantity, unit, and ingredient name. Ingredient weights are estimated based on the parsed quantity–unit information. In addition, each ingredient name is semantically mapped to the corresponding product category defined in the sustainability indicator databases, enabling the retrieval and computation of sustainability metrics.

We conduct human evaluation to assess the reliability of GPT outputs for these three tasks. Results show that 97%, 91%, and 94% of the samples are rated as acceptable for component phrase analysis, weight estimation, and sustainability indicator mapping, respectively. Additional details are provided in Appendix B.

## 5.3. User–Recipe Interaction Integration and Dataset Statistics

To facilitate personalized modeling, we align the constructed recipe dataset with user–recipe interaction data containing explicit ratings. These interactions are sourced from HUMMUS (Bölz et al., 2023) and the dataset introduced by (Majumder et al., 2019). We link user interactions to the specific recipes in our recipe dateset through name matching and metadata verification, resulting in a set of clean user–recipe–rating tuples.

The final dataset integrates recipes, standardized ingredients, multi-dimensional sustainability indicators, country

---

[2] https://www.ers.usda.gov/data-products/purchase-to-plate
[3] https://faunalytics.org/

labels, and user interactions into a unified dataset. Table 1 summarizes the key statistics of the dataset.

## 6. Empirical Evaluations

### 6.1. Experiment Settings

We evaluate the proposed framework from two aspects: how well the recommended recipes match user preferences, and how sustainable the recommended diets are across multiple dimensions, including nutritional adequacy, environmental impact, economic cost, and animal welfare. We consider a personalized top-$K$ recipe recommendation task. For each user, the model ranks a set of candidate recipes and returns the top-$K$ results. We report results for $K = 5, 10, 20$.

**Baselines.** Our task is closely related to general recommendation and food domain recommendation, requiring a diverse set of representative baseline methods. Accordingly, we use several widely adopted general recommendation models, including KNN(Wang et al., 2006), ICLRec(Chen et al., 2022), and MSSR(Lin et al., 2024), as general-purpose baselines. In addition, we use food recommendation methods, HAFR(Gao et al., 2020) and FGCN(Gao et al., 2022), that exploit food-related structures and relations as food-specific baselines. Furthermore, we employ GRAPE(Jing et al., 2025) as a sustainability-oriented recommendation baseline, which accounts for sustainability indicators in the recommendation process. All baseline methods are implemented using their original settings to ensure a fair comparison. Further details are provided in Appendix C.

**Evaluation metrics.** Cultural acceptability is captured through learned user preferences, which we evaluate using ranking-based metrics including NDCG@K and Recall@K. Higher values indicate better cultural acceptability. For the other sustainability dimensions (nutritional adequacy, environmental impact, economic cost, and animal welfare), we compute the average indicator values of the top-$K$ recommended recipes for each user. For these metrics, higher values indicate better outcomes for nutrition, while lower values indicate better outcomes for environment, price, and animal welfare.

### 6.2. Overall Results

Table 2 reports the overall performance of different methods under various Top-$K$ settings, evaluated in terms of recommendation accuracy and multiple sustainability indicators.

**Recommendation accuracy.** Our method achieves competitive NDCG and Recall compared with state-of-the-art baselines, obtaining the best or near-best results across several cutoff values. This demonstrates that our approach remains effective in recommending recipes that align well

*Table 2.* Performances of different methods for Top-K recommendation. The best results are bold, and the second-best are underlined. ↑ indicates that higher is better, ↓ indicates that lower is better. Origin refers to the indicator values computed from users' historically interacted recipes.

| Top-K | Metrics | Origin | KNN | ICLRec | MSSR | HAFR | FGCN | GRAPE | **Ours** |
|---|---|---|---|---|---|---|---|---|---|
| N=5 | NDCG | - | 0.0051 | 0.0119 | **0.0125** | 0.0058 | 0.0111 | 0.0105 | 0.0120 |
| | Recall | - | 0.0055 | 0.0162 | 0.0169 | 0.0075 | 0.0129 | 0.0152 | **0.0171** |
| | Nutrition ↑ | 38.7363 | 34.9750 | 34.5969 | 35.6965 | 34.5918 | 38.7869 | 39.5836 | **40.7221** |
| | Environment ↓ | 70.3760 | 67.6003 | 60.5654 | 63.4417 | **58.4069** | 62.7795 | 70.9837 | 69.9764 |
| | Economy ↓ | 7.3932 | 8.3118 | 4.9420 | 6.8017 | 6.2700 | 5.6742 | 5.2464 | **4.9258** |
| | Animal Welfare ↓ | 12.1631 | 16.1255 | 3.5907 | 9.7362 | 12.1456 | 6.3145 | 3.5327 | **2.7954** |
| N=10 | NDCG | - | 0.0056 | 0.0125 | 0.0133 | 0.0075 | 0.0129 | 0.0115 | **0.0136** |
| | Recall | - | 0.0075 | **0.0242** | 0.0211 | 0.0127 | 0.0186 | 0.0199 | 0.0223 |
| | Nutrition↑ | 38.7363 | 34.9288 | 35.1929 | 35.9286 | 35.6104 | 37.8545 | **39.5434** | 39.1429 |
| | Environment↓ | 70.3760 | 67.1324 | 59.8825 | 64.0748 | **59.8473** | 60.7506 | 69.5837 | 67.8675 |
| | Economy ↓ | 7.3932 | 8.3767 | 5.2925 | 6.8401 | 6.3471 | 5.7333 | 5.1543 | **4.8684** |
| | Animal Welfare ↓ | 12.1631 | 16.5685 | 4.4564 | 9.9498 | 11.9069 | 5.1571 | 5.2465 | **2.9186** |
| N=20 | NDCG | - | 0.0062 | 0.0170 | **0.0181** | 0.0098 | 0.0147 | 0.0131 | 0.0159 |
| | Recall | - | 0.0100 | **0.0340** | 0.0312 | 0.0212 | 0.0273 | 0.0265 | 0.0316 |
| | Nutrition↑ | 38.7363 | 34.9248 | 35.3663 | 35.9742 | 35.9525 | 37.1848 | 37.2563 | **37.5927** |
| | Environment↓ | 70.3760 | 66.1098 | **59.5418** | 64.1614 | 61.5087 | 64.2924 | 68.7837 | 66.2894 |
| | Economy ↓ | 7.3932 | 8.4511 | 5.6958 | 6.8710 | 6.4426 | 6.4037 | 6.8643 | **4.9445** |
| | Animal Welfare ↓ | 12.1631 | 17.0370 | 5.6385 | 10.3917 | 11.2196 | 7.8085 | 4.6847 | **3.0255** |

with user preferences.

**Sustainability performance.** Our method exhibits favorable results across multiple indicators. It consistently obtains high Nutrition scores while maintaining low Economy and Animal Welfare values, suggesting nutritionally improved, affordable, and welfare-aware recommendations. Although it does not achieve the lowest Environment score among all methods, we observe a common trade-off: recommendations with better nutritional quality often incur higher environmental impact. In contrast, our model produces a more balanced set of recommendations, avoiding extreme degradations in any single dimension. Moreover, it consistently outperforms historical user interactions across all sustainability dimensions.

**Additional LLM-based baselines.** We also compared against two additional LLM-based methods, Phi-3(Abdin et al., 2024) and KERL(Mohbat & Zaki, 2025), which are discussed in Appendix D.1. In brief, Phi-3 underperforms on all metrics, and while KERL achieves slightly higher recommendation accuracy, it falls significantly behind in environmental, economic, and animal welfare dimensions due to the lack of explicit constraint modeling.

**Sustainability prediction performance.** Table 3 presents the predictive performance of the multitask sustainability learning model across four sustainability dimensions. The model achieves low MAE and RMSE values together with consistently high $R^2$ scores, indicating accurate and stable prediction of sustainability indicators. These results demonstrate that the multitask learning formulation can reliably encode diverse sustainability attributes, providing

*Table 3.* Predictive performance of the multitask sustainability learning model across different dimensions.

| Task | MAE | RMSE | $R^2$ |
|---|---|---|---|
| Nutrition | 0.055 | 0.087 | 0.992 |
| Environment | 0.052 | 0.083 | 0.993 |
| Economy | 0.063 | 0.103 | 0.989 |
| Animal Welfare | 0.026 | 0.112 | 0.987 |

high-quality sustainability estimates that support the overall recommendation performance.

### 6.3. Ablation Study

We conduct ablation experiments to examine the contribution of the three core modules in the proposed framework. As shown in Table 4, removing the MoE-based *Recipe Sustainability Representation* leads to consistent degradation in sustainability indicators, while ranking accuracy is only slightly affected, highlighting the importance of explicitly encoding sustainability information at the recipe level. Removing the interest attention mechanism in the *Preference Learning* results in clear drops in both NDCG and Recall, accompanied by worse sustainability performance, indicating that modeling heterogeneous user interests is essential for both personalization quality and stable sustainability outcomes. Finally, when the *Personalized Constraint-aware Trade-off* is removed and recommendations are generated solely based on user preferences, sustainability indicators deteriorate substantially, demonstrating that preference-only recommendation tends to amplify unsustainable historical

*Table 4.* Ablation of MoE, interest attention, and constraint-aware trade-off modeling on Top-20 recommendation performance. MoE denotes the mixture-of-experts sustainability encoder in recipe representation; IntAttn denotes the interest attention mechanism in the preference learning module; Constraint denotes the personalized constraint-aware decision mechanism. Best results are highlighted in bold. ↑ indicates that higher is better, ↓ indicates that lower is better.

| MoE | IntAttn | Constraint | NDCG | Recall | Nutrition ↑ | Environment ↓ | Economy ↓ | Animal Welfare ↓ |
|---|---|---|---|---|---|---|---|---|
| × | ✓ | ✓ | 0.0155 | 0.0321 | 35.3628 | 68.6534 | 6.2455 | 4.9456 |
| ✓ | × | ✓ | 0.0110 | 0.0259 | 36.6532 | 68.2435 | 5.3234 | 4.3553 |
| ✓ | ✓ | × | **0.0173** | **0.0367** | 34.2454 | 70.8643 | 7.2345 | 5.9553 |
| ✓ | ✓ | ✓ | 0.0159 | 0.0316 | **37.5927** | **66.2894** | **4.9445** | **3.0255** |

*Table 5.* Item Coverage at different top-$k$ settings.

| Metric | History | Preference-only | Ours |
|---|---|---|---|
| IC@5 | 0.61 | 0.42 | 0.36 |
| IC@10 | 0.61 | 0.55 | 0.45 |
| IC@20 | 0.61 | 0.72 | 0.55 |

*Table 6.* Distribution of learned user-specific constraint thresholds $\boldsymbol{\lambda}_u$.

| | mean | std | P10 | P50 | P90 |
|---|---|---|---|---|---|
| $\lambda^{\text{Nutr.}}$ | 20.93 | 4.10 | 15.19 | 22.36 | 24.83 |
| $\lambda^{\text{Envi.}}$ | 67.25 | 47.40 | 46.11 | 63.63 | 92.83 |
| $\lambda^{\text{Econ.}}$ | 9.14 | 4.10 | 6.94 | 7.84 | 12.09 |
| $\lambda^{\text{Anim.}}$ | 9.37 | 20.80 | 3.80 | 5.10 | 13.65 |

*Table 7.* User case study comparing average sustainability indicators computed over historical interactions and Top-5 recommendations. ↑ indicates that higher is better, ↓ indicates that lower is better. Hist., Pref., and Ours denote the average indicators of the user's historical interactions, preference-only recommendations, and our recommendations, respectively. Nutr., Envi., Econ., and Anim. denote nutrition, environment, economy, and animal welfare, respectively.

| List | Nutr. ↑ | Envi. ↓ | Econ. ↓ | Anim. ↓ |
|---|---|---|---|---|
| Hist. | 37.51 | 86.78 | 9.97 | 18.70 |
| Pref. | 38.06 | 87.22 | 10.34 | 14.78 |
| **Ours** | **39.69** | **71.59** | **6.99** | **12.10** |

patterns, and that the proposed trade-off module is crucial for balancing user preferences with multi-dimensional sustainability constraints.

### 6.4. Diversity of recommendations.

We further evaluate recommendation diversity via Item Coverage (IC)@$k$, defined as the proportion of unique recipes recommended to all users among the top-$k$ results relative to the entire recipe set. Table 5 compares the coverage of the user interaction history, the preference-only model (i.e., our model without the constraint term), and our full method.

Incorporating sustainability constraints leads to a moderate reduction in coverage. However, our method maintains substantial coverage that remains comparable to the scale of observed user interactions. While the constraints prioritize more sustainable recipes, they do not strictly exclude less sustainable ones. Such items are typically ranked lower but may still be recommended.

### 6.5. Personalized Constraint Thresholds

To verify that the learned sustainability constraints are personalized, we analyze the distribution of user-specific constraint thresholds $\boldsymbol{\lambda}_u = [\lambda_u^{\text{Nutr.}}, \lambda_u^{\text{Envi.}}, \lambda_u^{\text{Econ.}}, \lambda_u^{\text{Anim.}}]$ across all users. Table 6 reports the mean, standard deviation, and

selected percentiles (P10, P50, P90) for each dimension.

The spread between percentiles (e.g., from P10 to P90) indicates that $\boldsymbol{\lambda}_u$ varies significantly across users for each sustainability dimension. For instance, the environmental constraint threshold ranges from 46.11 to 92.83, reflecting heterogeneous tolerance levels for environmental impact. These results demonstrate that the learned constraints are personalized and capture diverse user-specific sustainability boundaries.

### 6.6. User Case Study

To qualitatively examine the recommendation behavior of the proposed method, we present a case study on a representative user U1, which is shown in Table 7. Compared with both historical behavior and the preference-only baseline, our method substantially improves sustainability outcomes across all dimensions, while maintaining alignment with the user's original preferences.

## 7. Discussion

In this work, we reformulate personalized sustainable diet recommendation as a constraint-aware decision-making problem, addressing the gap between population-level sustainability guidance and individual dietary behavior. By modeling sustainability as learnable, user-specific constraints, our framework better reflects how individuals make real-world food choices. The proposed approach integrates

MoE-based recipe sustainability representation learning, multi-interest preference modeling, and personalized constraint learning within a unified optimization objective. Different from the works on AI-driven climate sustainability (Zhang et al., 2025; Bulian et al., 2024), this work addresses the domain of sustainable diets. It is also distinguished from sustainable food (Thomas et al., 2025) by its scope: whereas sustainable food focuses on the environmental integrity of production, a sustainable diet represents a consumer-centric paradigm that encompasses broader eating patterns. Experiments on the large-scale SusDiet dataset constructed in this work demonstrate that our method consistently promotes more sustainable recommendations while maintaining competitive personalization accuracy. These results highlight the potential of constraint-aware decision modeling as a principled foundation for sustainability-oriented recommendation systems and individualized dietary interventions.

Beyond the algorithmic performance, the broader significance of this framework lies in its potential to support more personalized and adaptive sustainable diet recommendation strategies. By modeling sustainability as a learnable user-specific constraint rather than a fixed global objective, the proposed approach enables recommendation systems to better accommodate heterogeneous dietary preferences while incorporating sustainability-oriented objectives. This formulation provides a possible pathway toward integrating sustainable considerations into everyday food recommendation scenarios. From a broader societal perspective, this work also provides a computational framework that may support future data-driven sustainable diet initiatives and public health strategies. Compared with traditional one-size-fits-all dietary guidelines, personalized recommendation frameworks offer the opportunity to account for differences in cultural background, dietary habits, and individual preference patterns when promoting sustainability-oriented food choices. In addition, the incorporation of sustainable indicators into recommendation modeling creates new possibilities for analyzing the interaction between dietary behavior and sustainability objectives at scale. Such capabilities may contribute to future research and applications related to sustainable consumption, precision nutrition, and environmentally informed dietary policy design.

**Limitations.** Despite the promising results, several limitations remain. First, the learned sustainability constraints are inferred from historical interaction data and may therefore only partially reflect users' true sustainability tolerance boundaries. Although the interaction data are derived from online recipe rating platforms rather than real-world purchase records and are thus less directly constrained by factors such as price or accessibility, exposure bias and unobserved socio-economic factors may still influence observed interactions. Future work could address these issues through exposure-aware recommendation strategies and richer contextual modeling. Second, the current framework also assumes static preferences and constraints, whereas both may evolve over time as users gain experience or awareness. In addition, while SusDiet provides a comprehensive set of sustainability indicators, these metrics are subject to uncertainty and simplification, which may propagate into the learned representations. Finally, our evaluation is limited to offline experiments, and the real-world behavioral impact of constraint-aware recommendations, such as long-term adherence, trust, and potential unintended effects, remains to be validated through online deployment or user-centered studies. Addressing these limitations will be crucial for further advancing constraint-aware decision modeling and realizing its full potential in supporting sustainable and responsible individual decision-making.

## Software and Data

The SusDiet dataset and code are available at `https://github.com/SustainFood/SusDiet`.

## Acknowledgements

This work was supported in part by the Beijing Natural Science Foundation (JQ24021) and the National Natural Science Foundation of China (62472411 and 62125207).

## Impact Statement

Our work is motivated by the broader challenges of sustainability, public health, and responsible consumption, with the goal of supporting more informed and personalized dietary decision-making. By promoting preference-aware food choices, this research explores pathways to facilitate the broader adoption of sustainable eating habits, which can collectively contribute to mitigating greenhouse gas emissions and land-use pressures while supporting human health. At the consumer level, the emphasis on gradual, adaptive adjustments respects individual autonomy and cultural practices, potentially reducing the resistance typically associated with rigid, one-size-fits-all interventions. Beyond individual choices, the insights derived from this multi-dimensional analysis can help policy-makers better understand how sustainability targets interact with heterogeneous population preferences, thereby supporting more adaptive, evidence-based policy design. At the same time, while data-driven consumption management can drive sustainability, it introduces vital concerns regarding fairness, transparency, and data governance. Managing these unintended societal effects requires careful oversight to ensure equitable transitions.

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

# A. Extended Methodology

## A.1. Recipe Sustainability Representation Module

In this module, we learn a structured and sustainability-aware representation of recipes by jointly modeling ingredient composition and multiple recipe-level sustainability attributes. This stage produces a recipe embedding that captures compositional semantics and serves as a reusable representation for downstream constraint-aware decision-making mechanism.

### A.1.1. INGREDIENT TOKENIZATION AND INPUT REPRESENTATION

Each recipe $r$ is described by a set of standardized ingredients and their corresponding quantities in grams, $\mathcal{G}_r = \{(i_j, q_j)\}_{j=1}^{N_r}$, where $i_j$ denotes a canonical ingredient token and $q_j \in \mathbb{R}_+$ denotes its quantity.

To enable neural sequence modeling, we construct a vocabulary over ingredient tokens by lowercasing all ingredient strings and filtering infrequent tokens below a minimum frequency threshold. Two special symbols are reserved to represent padding and unknown ingredients.

Since recipes contain a variable number of ingredients, each ingredient list is converted into a fixed-length sequence of length $L$ via truncation or padding. Specifically, ingredient tokens are mapped to integer indices, quantities are aligned accordingly, and a binary mask is generated to indicate valid ingredient positions. As a result, each recipe is represented by:

$$\mathbf{t}_r \in \mathbb{N}^L, \quad \mathbf{q}_r \in \mathbb{R}^L, \quad \mathbf{m}_r \in \{0,1\}^L,$$

where $\mathbf{m}_r[j] = 1$ if the $j$-th position corresponds to a real ingredient and $0$ otherwise. This formulation preserves both ingredient identity and quantitative contribution while allowing efficient mini-batch training.

### A.1.2. INGREDIENT–QUANTITY FUSION

To jointly encode ingredient identity and quantity information, we embed each ingredient token and project its corresponding quantity into a continuous space. Formally, for position $j$,

$$\mathbf{e}_{r,j} = \text{Embed}(\mathbf{t}_r[j]) \in \mathbb{R}^D,$$

$$\mathbf{u}_{r,j} = W_q \mathbf{q}_r[j] + b_q \in \mathbb{R}^{d_q}.$$

The two representations are concatenated and linearly projected to the model dimension:

$$\mathbf{x}_{r,j} = W_{\text{in}}[\mathbf{e}_{r,j}; \mathbf{u}_{r,j}] + b_{\text{in}} \in \mathbb{R}^D.$$

The resulting sequence $\mathbf{X}_r = [\mathbf{x}_{r,1}, \ldots, \mathbf{x}_{r,L}]$ serves as the input to the recipe encoder.

### A.1.3. MIXTURE-OF-EXPERTS RECIPE ENCODER

To capture heterogeneous compositional patterns across recipes, we adopt a Mixture-of-Experts (MoE) architecture consisting of $E = 4$ Transformer-based experts. Each expert independently encodes the ingredient sequence:

$$\mathbf{H}_r^{(e)} = \text{Transformer}^{(e)}(\mathbf{X}_r, \text{ padding\_mask} = (\mathbf{m}_r = 0)),$$

where padding positions are excluded from self-attention.

An expert-specific recipe representation is then obtained via masked mean pooling:

$$\mathbf{p}_r^{(e)} = \frac{\sum_{j=1}^{L} \mathbf{m}_r[j] \mathbf{H}_r^{(e)}[j]}{\sum_{j=1}^{L} \mathbf{m}_r[j] + \varepsilon} \in \mathbb{R}^D.$$

A gating network computes mixture weights based on a summary of the input sequence:

$$\bar{\mathbf{x}}_r = \frac{\sum_{j=1}^{L} \mathbf{m}_r[j] \mathbf{x}_{r,j}}{\sum_{j=1}^{L} \mathbf{m}_r[j] + \varepsilon},$$

$$\mathbf{g}_r = \mathrm{softmax}(W_2 \, \mathrm{ReLU}(W_1 \bar{\mathbf{x}}_r)) \,.$$

The final recipe embedding is computed as a gated combination of expert outputs:

$$\mathbf{z}_r = \mathrm{LayerNorm}\left( W_g \left( \sum_{e=1}^{4} \mathbf{g}_r[e]\mathbf{p}_r^{(e)} \right) + b_g \right) \in \mathbb{R}^D .$$

This MoE formulation enables different experts to specialize in distinct ingredient compositions while maintaining a unified representation space.

### A.1.4. MULTI-TASK SUSTAINABILITY PREDICTION HEADS

Given the recipe embedding $\mathbf{z}_r$, we jointly predict multiple sustainability-related attributes using task-specific regression heads. For each task $k \in \{1, \ldots, K\}$,

$$\hat{s}_r^{(k)} = W_{2k} \, \phi(W_{1k}\mathbf{z}_r + b_{1k}) + b_{2k},$$

where $\phi(\cdot)$ denotes a non-linear activation function. All tasks share the same recipe encoder but employ independent prediction heads, enabling shared representation learning with task-specific specialization. In our default setting, $K = 4$.

### A.1.5. TARGET TRANSFORMATION AND NORMALIZATION

The raw sustainability targets exhibit heavy-tailed distributions and heterogeneous scales. To stabilize optimization, we apply a sequence of transformations to each target value $s_r^{(k)}$. For sustainability dimensions where higher values are preferred, we multiply the corresponding scores by $-1$.

First, a signed logarithmic transformation compresses extreme values while preserving directionality:

$$s_r^{(k)} = \mathrm{sign}(s_r^{(k)}) \log(1 + |s_r^{(k)}|).$$

Second, transformed values are clipped using training-set quantiles to mitigate the influence of outliers. Finally, z-score normalization is applied:

$$s_r^{(k)} = \frac{s_r^{(k)} - \mu}{\sigma},$$

where $\mu$ and $\sigma$ are computed on training recipes only.

### A.1.6. LEARNED TASK-WEIGHTED MULTI-TASK OBJECTIVE

We train the recipe encoder and sustainability heads using a multi-task regression objective with learned task weighting. For a mini-batch of size B, the per-task mean squared error is computed as:

$$\ell_k = \frac{1}{B} \sum_{b=1}^{B} \left( \hat{s}_{r_b}^{(k)} - s_{r_b}^{(k)} \right)^2 .$$

Following uncertainty-based weighting, each task $k$ is associated with a learnable parameter $v_k$. The overall objective is:

$$\mathcal{L}_{\mathrm{task}} = \sum_{k=1}^{K} \left( \exp(-v_k) \ell_k + v_k \right).$$

This formulation allows the model to automatically balance gradients across tasks with different noise levels and scales, without manual tuning of task weights.

## A.2. Preference Learning Module via Multi-interest Mechanism

This module learns preference-oriented user and recipe representations from sequential interaction data for personalized recommendation. User dietary preferences are inherently evolutionary and dynamic. To capture such chronological patterns and multi-view culinary tastes, we adopt a multi-interest self-attention mechanism.

### A.2.1. USER-SIDE PREFERENCE ENCODING

For a user $u$, let $\mathcal{H}_u = \{(r_i, y_{u,r_i})\}_{i=1}^{N_u}$ denote the historical interaction set, where $r_i$ is an interacted recipe, $y_{u,r_i}$ is the observed rating, and $N_u$ is the maximum number of interactions.

**Pre-trained Structural Embedding Injection.**   Each historical recipe $r_i$ in the sequence is mapped into a continuous semantic space via the pre-trained embedding table $\mathbf{E}_u$:

$$\mathbf{e}_{r_i}^{(0)} = \mathbf{E}_u[r_i],$$

where $\mathbf{e}_{r_i}^{(0)} \in \mathbb{R}^d$.

**Sequential Token Construction.**   To capture the chronological progression of the user's dietary journey, we inject sequential order directly into the item state. We represent each step by combining the structural recipe vector with a learnable position bias token:

$$\mathbf{h}_i = \mathbf{e}_{r_i}^{(0)} + \mathbf{p}_i,$$

where $\mathbf{p}_i \in \mathbb{R}^d$ denotes the bias embedding for the $i$-th sequence position.

**Self-attention over user history.**   To model dependencies among interactions and extract multiple latent preference patterns, we apply self-attention on $\{\mathbf{h}_i\}_{i=1}^{N_u}$:

$$\mathbf{Q}_i^u = \mathbf{W}_Q^u \mathbf{h}_i, \quad \mathbf{K}_i^u = \mathbf{W}_K^u \mathbf{h}_i, \quad \mathbf{V}_i^u = \mathbf{W}_V^u \mathbf{h}_i,$$

$$a_{ij}^u = \frac{\exp\left(\frac{(\mathbf{Q}_i^u)^\top \mathbf{K}_j^u}{\sqrt{d}}\right)}{\sum_{k=1}^{N_u} \exp\left(\frac{(\mathbf{Q}_i^u)^\top \mathbf{K}_k^u}{\sqrt{d}}\right)}, \qquad \tilde{\mathbf{h}}_i = \sum_{j=1}^{N_u} a_{ij}^u \mathbf{V}_j^u.$$

### A.2.2. RECIPE-SIDE PREFERENCE REPRESENTATION

For a target candidate recipe $r$, we construct a multi-view preference profile by incorporating its main identity along with core cultural and composition dimensions.

We project the target recipe into its main ID space, standardized ingredient token space, and cultural region space via independent global lookup tables:

$$\mathbf{v}_{r,id} = \mathbf{E}_{\text{item}}[r], \qquad \mathbf{v}_{r,\text{ing}} = \mathbf{E}_{\text{ing}}[r], \qquad \mathbf{v}_{r,\text{reg}} = \mathbf{E}_{\text{reg}}[r],$$

where $\mathbf{E}_{\text{item}}$, $\mathbf{E}_{\text{ing}}$, and $\mathbf{E}_{\text{reg}}$ denote the learnable main identity, ingredient feature, and region feature embedding matrices, respectively, all mapping into $\mathbb{R}^d$. We also apply self-attention and pooling to model interactions among these embeddings to obtain $\tilde{\mathbf{v}}_{r,id}$, $\tilde{\mathbf{v}}_{r,\text{ing}}$ and $\tilde{\mathbf{v}}_{r,\text{reg}}$. Finally, the three representations are concatenated and projected through a linear transformation to obtain the final recipe representation $\mathbf{e}_r$.

### A.2.3. MULTI-INTEREST ATTENTION MECHANISM

Given user history tokens $\tilde{\mathbf{h}}_i$, we extract multi-faceted and context-aware latent preference patterns through a multi-layer Transformer block driven by multi-head self-attention.

We project the target recipe embedding as a query and the history tokens as keys and values:

$$\mathbf{Q}_r^c = \mathbf{W}_Q^c \tilde{\mathbf{v}}_{r,id}, \quad \mathbf{K}_i^c = \mathbf{W}_K^c \tilde{\mathbf{h}}_i, \quad \mathbf{V}_i^c = \mathbf{W}_V^c \tilde{\mathbf{h}}_i,$$

$$\beta_{i,r} = \frac{\exp\left(\frac{(\mathbf{Q}_r^c)^\top \mathbf{K}_i^c}{\sqrt{d}}\right)}{\sum_{j=1}^{N_u} \exp\left(\frac{(\mathbf{Q}_r^c)^\top \mathbf{K}_j^c}{\sqrt{d}}\right)}.$$

The user representation is:

$$\mathbf{e}_{u,id} = \sum_{i=1}^{N_u} \beta_{i,r} \mathbf{V}_i^c.$$

Similarly, $\tilde{\mathbf{v}}_{r,\text{ing}}$ and $\tilde{\mathbf{v}}_{r,\text{reg}}$ are processed through the same mechanism to obtain representation $\mathbf{e}_{u,\text{ing}}$ and representation $\mathbf{e}_{u,\text{reg}}$, respectively. Finally, the three representations are concatenated and projected through a linear transformation to obtain the final recipe representation $\mathbf{e}_u$.

### A.2.4. PREFERENCE SCORING

Finally, the preference score $\hat{y}_{u,r}$ for user $u$ and recipe $r$ is computed by:

$$\hat{y}_{u,r} = \langle \mathbf{e}_u, \boldsymbol{e}_r \rangle.$$

## B. Supplementary Information on Datasets

### B.1. Recipe Collection and Integration

We collected recipe data from multiple public sources to ensure diversity and broad cultural representativeness. Primary sources include:

- **recipeDB**(Batra et al., 2020): A large public recipe database scraped directly from its official website, containing 118,171 recipes, serving as the backbone of our corpus.

- **Yummly-28k**(Min et al., 2016) & **Yummly-66k**(Min et al., 2017): These popular platform datasets provide 27,638 and 66,615 recipes respectively, enriching recipe variety.

- **WorldCuisines**(Winata et al., 2025) & **World Wide Recipe**(Magomere et al., 2025): These smaller, focused datasets contain 2,414 and 765 recipes, with explicit country/region labels that enhance the cultural dimension of our data.

All selected datasets provide explicit country or region level labels, which allows cultural and geographic information to be retained as a specific attribute. Each recipe entry contains a recipe name, a country or regional tag, and an ingredient list expressed in form of natural language with associated quantity expressions. Unfortunately, these sources exhibit substantial heterogeneity in formatting conventions, ingredient naming practices, and unit systems. After merging all sources, we perform a systematic cleaning and deduplication process based on recipe titles, ingredient compositions, and metadata consistency. The resulting unified recipe corpus assigns a unique identifier to each recipe, which serves as the primary key for linking ingredient-level indicators and user interaction data in later stages.

### B.2. Ingredient Parsing and Standardization

The ingredient descriptions in the raw recipe data are expressed as unstructured text, often containing linguistic variations, modifiers, and ambiguous references, and the numerical representation is diverse. Direct string matching is therefore insufficient for reliably aligning such descriptions with structured ingredient databases required for sustainability analysis. To address this, we leverage GPT to assist in the accurate standardization required for sustainability analysis.

**Ingredient Phrase Parsing.** Firstly, we divide the description text of the ingredients into phrases. Following the strategy of (Zhang et al., 2024a), we employ GPT to parse each phrase into a set of structured words. Each phrase is decomposed into three components: a numerical quantity, a unit name, and an ingredient name (e.g., "1; cup; milk", "0.5; pound; pork"). Detailed prompts and parsing rules used for this step are provided in Figure 2.

**Ingredient Name Standardization.** The parsed ingredient names still contain numerous synonyms and variants (e.g., "tomato", "tomatoes", "cherry tomato"). We manually map them to a unified, canonical ingredient vocabulary. Through iterative cleaning and merging, we derived a final vocabulary of 7,340 distinct standardized ingredients.

**Sustainability indicator Mapping.** To enable the lookup and computation of sustainability-related indicators, each ingredient phrase is mapped to standardized ingredient categories defined by external authoritative databases. These databases cover multiple sustainability dimensions, including nutritional adequacy, environmental impact, economic cost, and animal welfare. Detailed descriptions of the databases for each sustainability dimension are provided as follows:

> **Prompt**
>
> Parse the ingredient phrase into JSON.
> Phrase: "{phrase}"
> Return exactly one JSON object in one line:
> {{"quantity":"","unit":"","ingredient_name":""}}
> Rules:
>   quantity: numeric only; empty if missing.
>   unit: unit word only; empty if missing.
>   ingredient_name: only the name of the ingredients, no quantity, no unit,
> other descriptive words for ingredients.
> If unsure, return empty strings. No explanation, no markdown, no extra text.

*Figure 2.* Prompt of component phrase analysis

- **nutritional adequacy.** We adopt standardized nutritional adequacy scores from (Clark et al., 2022), which provide nutrition indicators for 63 food product categories.

- **Environmental Impact.** Environmental indicators are derived from multiple life-cycle assessment studies. Specifically, (Poore & Nemecek, 2018) reports greenhouse gas emissions, land use, acidification, and eutrophication metrics for 43 food categories; (Clark et al., 2022) extends environmental impact estimates to 63 standardized food categories; and (Petersson et al., 2021) provides fine-grained carbon and water footprint data covering 324 and 320 food product categories, respectively. In our recommendation model, we use the environmental part of (Clark et al., 2022) as an environmental indicator.

- **Economic Cost.** Food price information is derived from the USDA Economic Research Service Purchase to Plate National Average Prices (PP-NAP) dataset[4], which reports retail-level cost estimates across 14,744 food product categories.

- **Animal Welfare.** Animal welfare indicators are obtained from Faunalytics[5], which provides welfare impact estimates covering 97 food product categories.

This semantic alignment is performed using GPT as a matching and disambiguation tool. Given an ingredient phrase and a candidate set of standardized categories, GPT identifies the most semantically appropriate match. In particular, due to the large number of categories in the economic cost database, we adopt a two-stage mapping strategy. Ingredients are first mapped to one of 197 coarse-grained categories, and subsequently assigned to a fine-grained category among the full set of 14,744 economic cost categories. Compared to rule-based or string-matching approaches, this strategy robustly handles lexical variability and ensures consistent alignment across heterogeneous data sources. Detailed prompts and parsing rules used in this step are provided in Figure 3.

### B.3. Unit Normalization and Quantity Conversion

A major challenge in aggregating ingredient-level sustainability indicators arises from the use of diverse and often informal quantity and unit expressions in recipes, ranging from volumetric units (e.g., cups, tablespoons) to qualitative descriptors (e.g., "a pinch", "a handful"). Moreover, many units appear in multiple lexical forms (e.g., lb vs. pound), further complicating quantitative analysis.

**Unit Unification.** We first normalized all parsed unit terms into 64 distinct standard units. For example, "tbsp", "T", and "tablespoons" are all mapped to "tablespoon".

**Quantity Estimation.** Subsequently, we convert all ingredient quantities into a unified measurement unit, "grams", to

---

[4]https://www.ers.usda.gov/data-products/purchase-to-plate
[5]https://faunalytics.org/

> **Prompt**
>
> You are a classifier. Given an ingredient text, choose the closest match from the provided lists.
> Ingredient: "{ing}"
> Choose one from Product options (or "none"): {prod_text}
> Return JSON exactly:
> {{"product": "<value from list or none>"}}

*Figure 3.* Prompt of sustainability indicator mapping

enable quantitative aggregation. Specifically, we construct unit–ingredient pairs from the previous stage, resulting in 20,105 distinct combinations. For each unit–ingredient pair, we apply GPT to estimate the corresponding mass in grams based on common culinary conventions and ingredient-specific density assumptions. The exact prompting strategy are detailed in Figure 4.

> **Prompt**
>
> You are estimating ingredient combo weights.
> Given a combo like "1 cup tomato" or "1 tsp salt", return JSON exactly:
> {{"weight_g": <number in grams>}}
> If unsure, return {{"weight_g": -1}}.
> Combo: "{combo}"

*Figure 4.* Prompt of quantity estimation

After this step, each recipe is represented as a set of ingredients with estimated masses in grams, providing a consistent and quantitative foundation for computing recipe-level sustainability indicators.

### B.4. Sustainability Metric Computation

With normalized ingredient representations, we compute recipe-level sustainability indicators by aggregating ingredient-level indicators obtained from the datasets described in Appendix B.2. Recipe-level indicators are computed as linear combinations of ingredient-level indicators weighted by ingredient mass. As a result, each recipe is associated with a comprehensive vector of sustainability indicators spanning all dimensions.

### B.5. Geographic and cultural lable standardization

Country and region tags associated with recipes were standardized by mapping them to a unified taxonomy aligned with the United Nations M49 standard[6].

Recipes were ultimately tagged with one of 131 distinct country labels.

---

[6] https://unstats.un.org/unsd/methodology/m49/

*Table 8.* Statistics of the constructed user–recipe interaction dataset

| Dataset | Recipe | User | Interaction |
|---|---|---|---|
| (Bölz et al., 2023) | 507,335 | 302,412 | 1.916,424 |
| (Majumder et al., 2019) | 231,637 | 226,570 | 1,132,367 |
| SusDiet(Ours) | 149,036 | 179,869 | 744,138 |

### B.6. User–Recipe Interaction Alignment

To support personalized recommendation and preference modeling, we align the constructed recipe dataset with user–recipe interaction data. We utilize interaction from the HUMMUS(Bölz et al., 2023) dataset and an additional dataset reported in (Majumder et al., 2019), which contain 1,916,424 and 1,132,367 raw interaction records, respectively. Each record includes a username, a recipe name, and an explicit rating.

We link these interactions to our constructed recipe corpus through recipe name matching and metadata verification. After alignment and filtering, we retain 744,138 valid interaction records. Each interaction is represented as a tuple indicating a user-assigned rating for a specific recipe, linked via the shared recipe identifier. The statistical results are shown in Table 8.

### B.7. Human Evaluation of GPT-Based Processing

All GPT-assisted steps, including component phrase analysis, sustainability indicator mapping, quantity estimation, are subject to human evaluation. For each step, we randomly sampled 500 instances and asked human annotators to score the GPT output based on correctness and semantic sufficiency, with a score range of 1-4 for each task. The resulting scores are used to assess the reliability of GPT-based preprocessing.

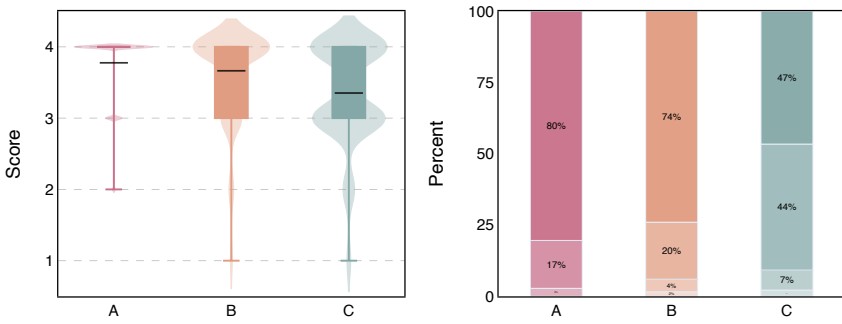

*Figure 5.* Human evaluation results of GPT performance across (A) Component Phrase Analysis, (B) Sustainability Indicator Mapping, and (C) Quantity Estimation. Left: Score distributions for each task, with the horizontal line denoting the mean human score. Right: Percentage composition of scores for each task.

The experimental results demonstrate that the GPT achieves consistently strong performance across the three tasks, with average human evaluation scores of 3.776, 3.664, and 3.352, respectively. The score distributions aggregated from all evaluators are shown in Figure 5. Across the three tasks, 97%, 94%, and 91% of the samples were rated as acceptable (scores between 3 and 4), indicating that the majority of GPT generated outputs meet practical quality requirements. These results suggest that the GPT can reliably produce usable outputs for component phrase analysis, sustainability indicator mapping, and quantity estimation.

Furthermore, we compared two alternative approaches for ingredient weight estimation: (1) providing the raw ingredient phrase directly to GPT, and (2) a two-stage approach in which the phrase is first parsed into quantity, unit, and ingredient components before weight estimation. We conducted a manual evaluation using a 4-point scale on 200 samples for each approach, resulting in average scores of 2.85 and 3.36, respectively. The results indicate that the raw ingredient phrases often contain extraneous or conflated information, which negatively impacts the GPT's weight estimation performance. Therefore, we adopted the two-stage parsing-and-estimation approach for the final method.

### B.7.1. HUMAN EVALUATION CRITERIA

For the component phrase analysis task, a score of 4 indicates that all three elements (quantity, unit, and ingredient phrase) are correctly identified with accurate boundaries. A score of 3 denotes that the core information of all three elements is correct, though minor presentational imperfections may exist. A score of 2 reflects that at least one element is correctly identified, while the others contain notable errors. A score of 1 signifies that most or all information is incorrect, resulting in a failure to properly segment any of the elements or in incoherent or irrelevant output.

For the sustainability indicator mapping task, a score of 4 represents an exact match to an item in the reference list or a match to a standardized synonym. A score of 3 corresponds to a match to the correct broader category, with minor differences that do not affect practical usability. A score of 2 indicates a match to a related but not entirely correct item, where clear errors are present yet remain within the same general category. A score of 1 is assigned when the match is to a completely unrelated category or when the matching attempt fails entirely.

For the quantity estimation task, a score of 4 is given when the numerical estimate is either exact or falls within a reasonable standard range, consistent with common sense and culinary norms. A score of 3 is assigned for estimates with minor deviation that remain within an acceptable margin. A score of 2 reflects estimates with substantial deviation from the expected value, while a score of 1 is reserved for estimates with severe deviation that clearly contradict common sense.

### B.8. Dataset Analysis

We further conduct exploratory analyses to characterize the dataset in terms of ingredient distributions, sustainability indicator ranges, and geographic coverage. As shown in Figure 7, the ingredient frequency distribution exhibits a clear long-tail structure: a limited set of widely used ingredients appears with high frequency, while a large number of ingredients occur sparsely. This pattern reflects realistic dietary practices and ensures strong coverage of common food items.

As illustrated in Figure 6, the dataset demonstrates broad geographic coverage, with recipes spanning a diverse set of countries and regions, albeit with uneven representation across cuisines. Beyond structural statistics, the dataset integrates multiple dimensions of sustainability indicators, including environmental impact, nutritional adequacy, economic cost, and animal welfare. These indicators exhibit substantial variation across recipes and regions, highlighting diverse sustainability trade-offs present in real-world dietary choices. Collectively, these characteristics indicate that the dataset captures rich diversity across ingredients, cultures, and sustainability dimensions, while also revealing potential imbalances that should be considered in the design and evaluation of downstream modeling tasks.

## C. Experimental Setup and Implementation Details

### C.1. Implementation Details

In our experiment, we select the learning rate from the set $\{0.0005, 0.001, 0.005\}$, the embedding size $d$ from $\{64, 128, 256\}$, and the batch size from $\{64, 128, 256\}$. We tune the number of attention heads from $\{2, 4, 8\}$. For the loss function trade-offs, we select hyperparameter $\alpha = 0.5$ the ranking weight $\beta_1$ from $[0.1, 1.0]$, the sustainability penalty weight $\beta_2$ from $[0.1, 2.0]$, and the L2 regularization coefficient $\beta_3$ from $[10^{-6}, 10^{-4}]$.

### C.2. Baselines

We now describe the baseline methods used to evaluate the proposed model.

- **KNN**(Wang et al., 2006): This method reformulates memory-based collaborative filtering within a probabilistic framework that fuses user-based and item-based similarities, treating each observed rating as a weighted predictor of missing ratings. By jointly leveraging similarities across users and items with background smoothing, it improves robustness to data sparsity compared to traditional neighborhood-based approaches.

- **ICLRec**(Chen et al., 2022): This approach introduces a latent intent variable for sequential recommendation and learns user intent distributions from unlabeled interaction sequences via clustering, which are integrated into recommendation through intent-aware contrastive self-supervised learning.

- **MSSR**(Lin et al., 2024): This method models user behavior by jointly learning representations from item sequences and multiple side-information sequences using a multi-sequence integrated attention mechanism that captures both

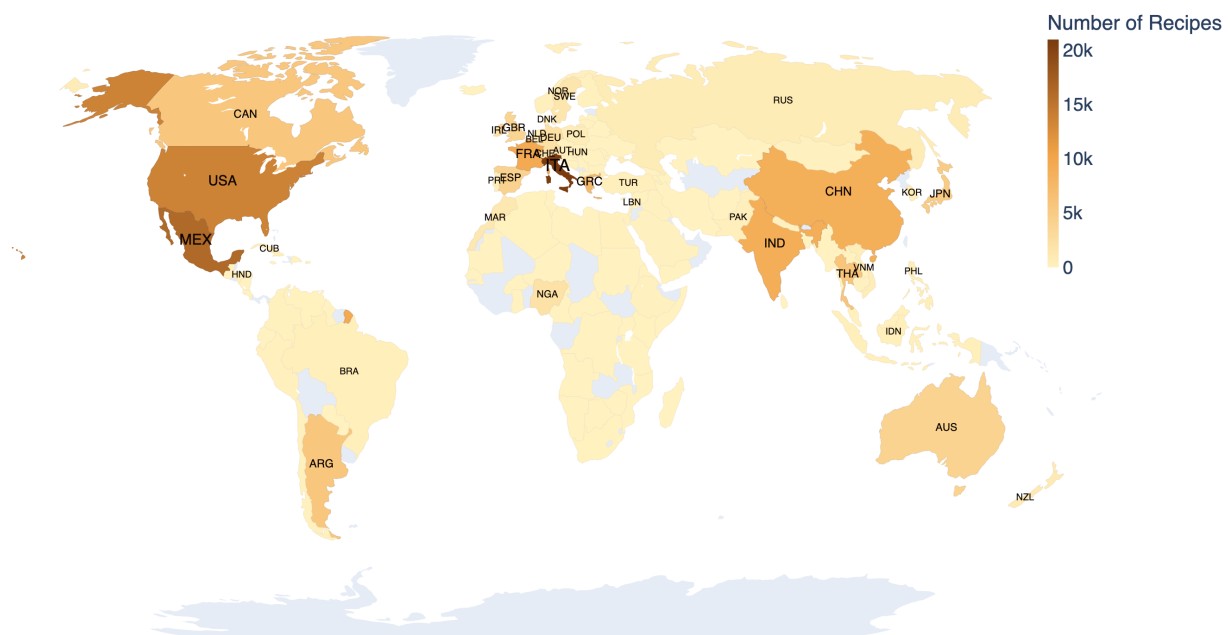

*Figure 6.* Statistical map of the distribution of the number of recipes in different countries

intra- and inter-sequence dependencies. It further aligns heterogeneous user representations via contrastive learning and incorporates side information of candidate items for more comprehensive preference modeling.

- **HAFR**(Gao et al., 2020): This approach employs a hierarchical attention network to jointly model user–recipe interactions and multimodal recipe content. Ingredient-level and component-level attentions adaptively fuse different modalities conditioned on user preferences.

- **FGCN**(Gao et al., 2022): This method models food recommendation by constructing a heterogeneous graph over ingredient–ingredient, ingredient–recipe, and recipe–user relations, and applies multi-layer graph convolution with information propagation to capture high-order connectivity and enhance user and recipe representations.

- **GRAPE**(Jing et al., 2025): This approach models green food recommendation by jointly capturing users' sequential preferences over items and multiple sustainability indicators through integrated self- and cross-attention, and introduces dedicated green loss functions to balance recommendation accuracy with the promotion of more sustainable food choices.

## D. Additional Experimental Results

### D.1. Additional LLM-based Baselines

To further evaluate LLM-based methods on our dataset, we constructed the dataset following a pipeline inspired by prior work: (i) recipes, ingredients, and user interactions were processed to form structured knowledge graphs and user behavior sequences; (ii) each recipe graph contains title, ingredients, nutrition information, and tags derived from country; (iii) from user sequences, we generated QA-style samples with constraints on sustainability attributes.

We evaluated both Phi-3-mini-128k (Abdin et al., 2024) and the KERL model (Mohbat & Zaki, 2025) on this dataset. Preliminary results at Top-10 recommendations are shown in Table 9.

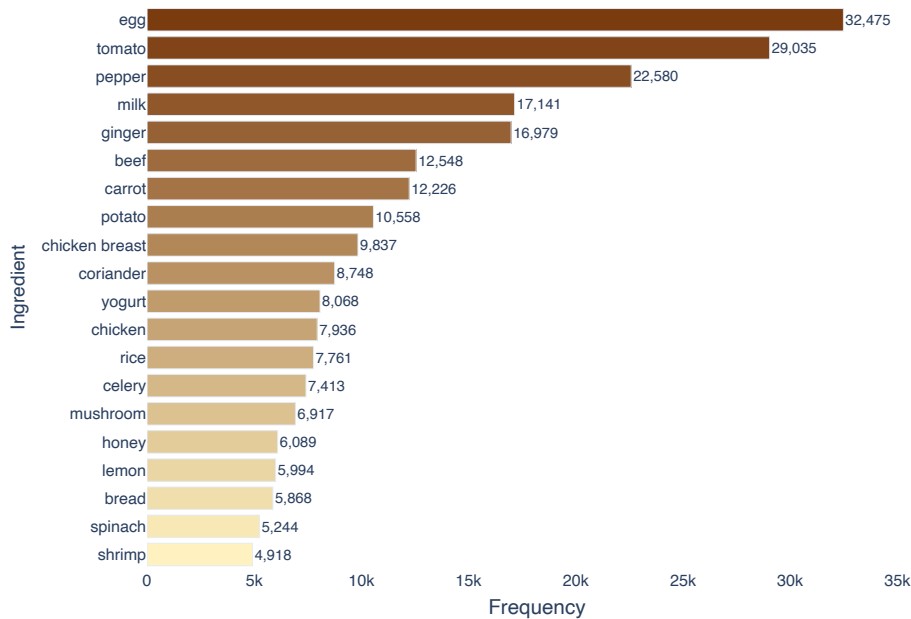

*Figure 7.* Frequency Distribution of the Top 20 Most Common Ingredients in the Recipe Dataset

*Table 9.* Performance of additional LLM-based baselines (Top-10).

| Metric | Phi-3 | KERL | Ours |
|---|---|---|---|
| NDCG@10 | 0.0035 | 0.0160 | 0.0136 |
| Recall@10 | 0.0087 | 0.0243 | 0.0223 |
| Nutrition@10 ↑ | 48.5782 | 43.5833 | 39.1429 |
| Environment@10 ↓ | 125.0952 | 76.8827 | 67.8675 |
| Economy@10 ↓ | 7.5879 | 9.5769 | 4.8684 |
| Animal Welfare@10 ↓ | 4.3617 | 20.1999 | 2.9186 |

As shown, Phi-3 performs poorly on recommendation metrics and struggles to balance user preferences with multiple sustainability dimensions, likely due to its limited understanding of these aspects. KERL slightly outperforms our method on NDCG@10 and Recall@10, but falls significantly behind on Environment, Economy, and Animal Welfare metrics. This may be because KERL does not enforce strict constraints, instead allowing greater trade-offs in favor of user preference dimensions. KERL performs relatively well on Nutrition, likely because it was originally trained on datasets containing nutritional information. Note that due to time and computational constraints, these models were used without fine-tuning.

## D.2. Comparing with LLM

We conducted experiments using Llama 3-8B to generate sustainability representations for all recipes. The generated representations were used as item-side features in most baselines (ICLRec, MSSR, HAFR, FGCN, GRAPE) and also in our framework (replacing the sustainability representation stage while keeping the constraint mechanism). Results on Top-20 recommendations are shown in Table 10.

The results show that our method consistently outperforms all LLM-based variants in recommendation accuracy and sustainability performance, indicating that LLMs alone are not sufficient for this task. When directly using LLMs for sustainability representation (Ours(LLM)), direct LLM outputs lack structured and reliable modeling, and cannot explicitly capture fine-grained sustainability dimensions in a single step, leading to worse performance than our method. This

*Table 10.* Top-20 recommendation performance of different methods using LLM for sustainability representations. The best results are in bold. ↑ indicates that higher is better, ↓ indicates that lower is better.

| Metrics | ICLRec(LLM) | MSSR(LLM) | HAFR(LLM) | FGCN(LLM) | GRAPE(LLM) | Ours(LLM) | **Ours** |
|---|---|---|---|---|---|---|---|
| NDCG | 0.0124 | 0.0154 | 0.0092 | 0.0105 | 0.0137 | 0.0148 | 0.0159 |
| Recall | 0.0248 | 0.0201 | 0.0159 | 0.0224 | 0.0264 | 0.0293 | 0.0316 |
| Nutrition ↑ | 33.6432 | 36.0231 | 36.6532 | 32.6542 | 36.4352 | 37.1464 | 37.5927 |
| Environment ↓ | 63.3256 | 66.7433 | 64.6534 | 71.5634 | 68.6534 | 67.6534 | 66.2894 |
| Economy ↓ | 6.6743 | 7.0234 | 7.7643 | 8.4534 | 6.4764 | 6.3462 | 4.9445 |
| Animal Welfare ↓ | 10.4234 | 9.3454 | 12.3244 | 12.2345 | 8.9753 | 5.7434 | 3.0255 |

observation is consistent with Appendix B.7: in the data construction stage, we compared direct LLM generation with a structured two-stage approach, where the latter significantly outperformed the former (3.36 vs. 2.85 in human evaluation). This demonstrates that LLM outputs are inherently less reliable without additional learning. Besides, replacing the sustainability representations in conventional baselines with LLM-generated features (baseline(LLM)) does not consistently improve performance and even degrades some metrics, further indicating that LLMs are not well-suited for learning reliable sustainability representations alone.

### D.3. Ablation Study on the Sustainability Dimensions

To further demonstrate the necessity of modeling each sustainability dimension as an explicit constraint, we conduct ablation studies by removing the constraint for each dimension individually while keeping the rest of the framework unchanged. The results on Top-20 recommendations are summarized below. The results show that removing any single constraint consistently degrades performance on the corresponding sustainability dimension, and in some cases also negatively affects other dimensions. In contrast, the full model achieves the best overall balance across all dimensions.

*Table 11.* Ablation study on the sustainability dimensions. ↑/↓ indicate higher/lower is better.

| | Nutr.↑ | Envi.↓ | Econ.↓ | Anim.↓ |
|---|---|---|---|---|
| w/o Nutr. | 35.3454 | 66.1432 | 4.9464 | 3.1235 |
| w/o Envi. | 37.6464 | 70.0443 | 6.5433 | 4.7532 |
| w/o Econ. | 36.1467 | 67.4213 | 7.0745 | 4.9742 |
| w/o Anim. | 37.4642 | 68.6434 | 6.3134 | 8.4523 |
| Full Model | 37.5927 | 66.2894 | 4.9445 | 3.0255 |

### D.4. Analysis on High-Conflict Scenarios

To address concerns regarding high-conflict scenarios, we performed a quantitative boundary test by stratifying users into Top 10% (least sustainable / high-conflict) and Bottom 10% (most sustainable) groups based on their historical environmental scores. Table 12 reports the recommendation accuracy and sustainability indicators for each group under the preference-only baseline and our full model.

*Table 12.* Performance on high-conflict (Top 10%) and low-conflict (Bottom 10%) user groups. History indicators are computed from users' historically interacted recipes.

| Metric | History | | Preference-only | | Ours | |
|---|---|---|---|---|---|---|
| | Top 10% | Bottom 10% | Top 10% | Bottom 10% | Top 10% | Bottom 10% |
| NDCG@10 | – | – | 0.0123 | 0.0142 | 0.0122 | 0.0135 |
| Recall@10 | – | – | 0.0256 | 0.0256 | 0.0230 | 0.0244 |
| Nutrition@10 | 57.2219 | 22.4635 | 37.0862 | 34.5305 | 40.7569 | 35.3851 |
| Environment@10 | 164.5703 | 19.6472 | 71.4178 | 54.7939 | 69.5293 | 46.2580 |
| Economy@10 | 9.9947 | 4.3291 | 7.0616 | 6.1799 | 5.1901 | 4.0410 |
| Animal Welfare@10 | 13.8657 | 5.1481 | 9.5334 | 7.3426 | 3.6893 | 2.3606 |

For the Top 10% group, whose preferences are most opposed to sustainability requirements, our model achieved improve-

ments across all four sustainability dimensions compared to the preference-only baseline, with only a marginal loss in recommendation accuracy. Furthermore, the Bottom 10% group exhibited even greater gains in sustainability indicators. This demonstrates that the sustainability constraints remain effective even at extreme boundaries, allowing the framework to nudge the most unsustainable users toward better alternatives without significantly sacrificing accuracy. These findings confirm the robustness and effectiveness of our approach in resolving high-conflict dietary trade-offs.

### D.5. User Case Study

*Table 13.* Detailed recipe-level results for User U1 with pork- and beef-oriented preferences. ↑/↓ indicate higher/lower is better.

| Method | Recipe | Nutrition ↑ | Environment ↓ | Economy ↓ | Animal Welfare ↓ |
|---|---|---|---|---|---|
| *Historical interactions* | | | | | |
| History | Spicy Beef Noodle Soup | 39.23 | 102.88 | 13.83 | 30.90 |
| History | Mapo Tofu With Crispy Chinese Sausage | 45.46 | 64.26 | 14.65 | 21.13 |
| History | Kung Pao Chicken | 35.14 | 89.33 | 7.63 | 12.43 |
| History | Braised Pork Rice | 38.91 | 106.96 | 9.32 | 18.88 |
| History | Fried Rice w/ Egg & Ham | 28.80 | 70.47 | 4.41 | 10.17 |
| *Preference-only recommendations (Top-5)* | | | | | |
| Preference-only | Double-Cooked Pork | 43.65 | 72.42 | 5.93 | 10.45 |
| Preference-only | Beef Pepper Stir-fry | 50.21 | 108.13 | 8.42 | 15.34 |
| Preference-only | Hotpot Lamb Slices | 26.11 | 124.70 | 16.61 | 20.42 |
| Preference-only | Crispy Fried Chicken | 32.48 | 62.21 | 10.80 | 14.91 |
| Preference-only | Pork Ribs Soup | 37.83 | 68.68 | 9.95 | 12.77 |
| *Our recommendations (Top-5)* | | | | | |
| Ours | Spicy Beef Noodle Soup (Lean Cut) | 28.82 | 61.22 | 5.65 | 16.25 |
| Ours | Chili Garlic Pork with Eggplant (Reduced Oil) | 45.84 | 69.37 | 4.26 | 7.10 |
| Ours | Dan Dan Noodles with Lean Pork | 33.92 | 107.84 | 5.18 | 6.43 |
| Ours | Stir-fry Chicken & Broccoli | 39.61 | 73.90 | 15.82 | 18.24 |
| Ours | Tomato Beef & Tofu Stew (Light) | 50.24 | 45.60 | 4.01 | 12.49 |

