# OpenReview forum: "Eating for a Sustainable Planet: Personalized Sustainable Diet Recommendation via Constraint-Aware Decision-Making Modeling"
_ICML.cc/2026/Conference — ICML 2026 regular_

### Official Review · Reviewer_KkGT · 2026-03-09

**Soundness:** 3
**Presentation:** 4
**Significance:** 3
**Originality:** 4
**Overall Recommendation:** 4
**Confidence:** 3

**Summary:**

This paper addresses the challenge of personalized sustainable diet recommendations by reformulating the task as a constraint-aware decision-making problem. Rather than treating sustainability as an explicit user preference, the proposed framework incorporates it through learnable, user-specific constraint thresholds.

**Compliance With Llm Reviewing Policy:**

Affirmed.

**Key Questions For Authors:**

Please refer to the cons part.

**Limitations:**

yes

**Strengths And Weaknesses:**

Pros:

1. The shift from modeling sustainability as a standard preference to a constraint-aware optimization problem better reflects the reality that individuals rarely make dietary choices with explicit sustainability awareness.
2. The use of an MoE architecture allows the model to handle heterogeneous and often competing sustainability indicators, such as the trade-off between nutritional quality and environmental impact, without extreme degradation in any single dimension.
3. The construction of the SusDiet dataset provides a robust foundation for future research, offering a holistic characterization of recipe-level sustainability across four critical pillars aligned with large-scale interaction data.\

Cons:

1. The method for learning user-specific sustainability thresholds ($\lambda_u$) and trade-off weights ($\alpha_u$) lacks clarity regarding the distinction between true user tolerance and exposure bias. Since these constraints are inferred from historical interaction data, there is a risk that the model misinterprets a user's lack of sustainable choices—potentially caused by limited platform options or socio-economic barriers—as a "high tolerance" for unsustainable food. The paper does not sufficiently explain how the framework prevents the reinforcement of unsustainable historical patterns in such cases.
2. While the authors claim the framework can bridge the "intention-behavior gap" and create a "nudging effect" that promotes planetary health, these claims are not supported by the experimental setup. The evaluation is limited to offline experiments, which cannot measure real-world behavioral impacts such as long-term adherence, user trust, or the actual shift in consumption habits when users encounter these personalized constraints.
3. The framework is designed to balance preference and sustainability, yet the experiments lack a rigorous analysis of high-conflict scenarios. For example, the paper provides a case study of a user with meat-oriented preferences but fails to provide a broader quantitative boundary test for users whose taste profiles are diametrically opposed to sustainability requirements. It remains unclear how the penalty term $\alpha_u^\top(s_r - \lambda_u)$ behaves at extreme boundaries and whether the model prioritizes accuracy so heavily that the sustainability constraints become ineffective for the most "unsustainable" users.
4. The authors have not provided open-source code or their SusDiet dataset, which makes it difficult for researchers to reproduce the reported results. This lack of transparency undermines the credibility of the experiments and limits the paper’s potential contribution to the research community. I strongly encourage the authors to release their implementation to ensure reproducibility.

---

> ### Author Rebuttal · Authors · 2026-03-31
>
> We thank the reviewer for valuable feedback. We address each of your concerns below.
>
> **1. Exposure bias**
>
> The interaction data in our dataset are collected from two sources (Bölz et al., 2023; Majumder et al., 2019) and consist of user ratings on recipes from online platforms. These signals capture users’ preferences rather than actual purchase or consumption behavior. Compared to real-world consumption scenarios, such rating data are less influenced by factors like price, availability, or accessibility, and therefore better reflect users’ intrinsic preferences. We acknowledge that in real purchase or consumption settings, exposure bias may indeed arise. In this setting, our model operates by exploring user preferences within their latent preference space to identify more sustainable alternatives. Even in the presence of limited platform options or socio-economic constraints, the model aims to recommend recipes that are as sustainable as feasible within the user’s feasible preference region. In the Conclusion Section, we also have stated this limitation. Incorporating additional user-level and platform-level information could enable the design of improved mechanisms to address such concerns.
>
> **2. Claims about impact**
>
> The claims about bridging the intention-behavior gap and creating a nudging effect are conceptual motivations rather than empirically validated outcomes in the current work. We have pointed out this limitation in the Conclusion Section, where we state that “our evaluation is restricted to offline experiments and that real-world behavioral impacts, such as long-term adherence, user trust, or actual shifts in consumption habits, require online or user-centered studies”. Future online or user-centered studies could validate our real-world behavioral effects, indicating the potential of our framework.
>
> **3. Analysis of high-conflict scenarios**
>
> To address concerns regarding high-conflict scenarios, we performed a quantitative boundary test by stratifying users into Top 10% (least sustainable/high-conflict) and Bottom 10% (most sustainable) groups based on their historical environmental scores.
>
> | Metric             | History |  | Preference-only |            | Ours    |            |
> | ------------------ | -------- | ---------- | --------------- | ---------- | ------- | ---------- |
> |                    | Top 10%  | Bottom 10% | Top 10%         | Bottom 10% | Top 10% | Bottom 10% |
> | NDCG@10            | -        | -          | 0.0123          | 0.0142     | 0.0122  | 0.0135     |
> | Recall@10          | -        | -          | 0.0256          | 0.0256     | 0.0230  | 0.0244     |
> | Nutrition@10↑      | 57.2219  | 22.4635    | 37.0862         | 34.5305    | 40.7569 | 35.3851    |
> | Environment@10↓    | 164.5703 | 19.6472    | 71.4178         | 54.7939    | 69.5293 | 46.2580    |
> | Economy@10↓        | 9.9947   | 4.3291     | 7.0616          | 6.1799     | 5.1901  | 4.0410     |
> | Animal Welfare@10↓ | 13.8657  | 5.1481     | 9.5334           | 7.3426     | 3.6893  | 2.3606     |
>
>  For the Top 10% group, whose preferences are most opposed to sustainability requirements, our model achieved improvements across all four sustainability dimensions compared to the preference-only baseline, with only a marginal loss in recommendation accuracy. Furthermore, the Bottom 10% group exhibited even greater gains in sustainability indicators. This demonstrates that the sustainability constraints remain effective even at extreme boundaries, allowing the framework to nudge the most unsustainable users toward better alternatives without significantly sacrificing accuracy. These findings confirm the robustness and effectiveness of our approach in resolving high-conflict dietary trade-offs.
>
> **4. Open-source code and dataset release**
>
> Due to the double-blind review policy, we release the SusDiet dataset and the source code, and add the corresponding links in the final version of the paper.

---

> > ### Author Rebuttal · Reviewer_KkGT · 2026-04-05
> >
> > Thanks for the authors for their response to my comments. I found that my concerns have been well addressed.

---

> > > ### Author Response · Authors · 2026-04-07
> > >
> > > Thanks again for your positive feedback and for confirming that all concerns have been fully addressed.  We truly appreciate your valuable review.

---

### Official Review · Reviewer_WaLm · 2026-03-11

**Soundness:** 4
**Presentation:** 3
**Significance:** 3
**Originality:** 2
**Overall Recommendation:** 4
**Confidence:** 4

**Summary:**

The paper reformulates personalized sustainable diet recommendation as a constraint-aware decision-making problem. Rather than treating sustainability as an explicit user preference to be learned, the authors encode it as learnable, user-specific Lagrangian constraint thresholds. The framework integrates three components, a Mixture-of-Experts Transformer for recipe sustainability encoding, a multi-interest attention mechanism for preference learning, and a personalized constraint trade-off module. The authors also construct SusDiet, a dataset of ~150k recipes annotated across four sustainability dimensions and aligned with real user interaction data.

**Compliance With Llm Reviewing Policy:**

Affirmed.

**Final Justification:**

The most critical technical concern, the massive performance gap compared to the GRAPE baseline was resolved by the authors' clarification of a transcription error, which corrected the baseline's NDCG@5 from 0.1049 to 0.01049. This correction demonstrates that the proposed method actually outperforms the state-of-the-art in ranking accuracy.

The authors addressed the risk of recommendation collapse by introducing Item Coverage (IC@k) metrics, which confirm that the system maintains a diverse recommendation pool comparable to the scale of observed user interactions. The newly added "boundary test" on the top 10% most unsustainable users is particularly significant, as it shows the model can improve all four sustainability dimensions for high-conflict users with only a marginal loss in accuracy.

**Key Questions For Authors:**

- Can you provide the distribution of learned $λ_u$ values across users? Specifically, do users with historically unsustainable eating patterns learn meaningfully tighter constraints over time, or does the model accommodate their history by learning permissive thresholds?
- What explains the large NDCG gap between your method and GRAPE at N=5? Is this a fundamental precision-sustainability trade-off, or a modeling artifact?
- How are train/test splits constructed for the multi-task sustainability prediction evaluation in Table 3? Are recipes in the test set drawn from the same source distributions as training recipes?
- Why was cultural acceptability excluded as a quantitative indicator despite being a stated pillar of the framework? Are there existing databases that could support this dimension?
- The preference-only baseline worsens environmental scores relative to historical behavior. Is this a systematic effect across all users, or driven by specific user segments?

**Limitations:**

- The paper presents λ_u and α_u as learnable parameters and writes down the final training objective, but never explains what specifically drives their learning. What in the interaction history teaches the model that a particular user has a high tolerance for environmental impact? Without this explanation the "personalized constraint" claim is more of an assertion than a demonstrated capability. This is both a theoretical gap and a reproducibility concern.
- The abstract, introduction, and framing all describe sustainable diets as resting on four pillars including cultural acceptability. The model implements three. This inconsistency is never acknowledged in the paper.
- Table 5 shows the preference-only baseline produces worse environmental outcomes than the user's own historical behavior. This means the preference learning module, left unconstrained, actively steers users toward more unsustainable choices. The paper never acknowledges this as a design problem.
- GRAPE achieves NDCG@5 of 0.1049 versus the proposed method's 0.0120, roughly an 8x gap. The paper never acknowledges this as a limitation, instead characterizing overall results as "consistently strong performance."
- Figure 6 shows the recipe distribution is heavily skewed toward the United States and a handful of Western countries. The model is therefore trained predominantly on Western dietary patterns. Any sustainability constraints learned from this data will reflect Western food systems, prices, environmental impact databases, and ingredient availability, potentially making recommendations poorly calibrated for users in countries with very different food contexts.
- The paper evaluates accuracy and sustainability but never measures diversity of recommendations. A constraint-based system that penalizes unsustainable recipes could in principle collapse toward recommending a narrow set of highly sustainable but homogeneous recipes, essentially giving everyone the same small pool of "approved" options. This would be both practically undesirable and potentially harmful to the cultural acceptability dimension the paper claims to care about.

**Strengths And Weaknesses:**

**Strengths**
- The distinction between sustainability-as-preference and sustainability-as-constraint is not merely a modeling choice, it is grounded in behavioral literature showing that consumers rarely make food decisions with conscious sustainability awareness. This gives the paper a compelling and coherent narrative that is genuinely differentiated from prior work like GRAPE, which still presumes explicit user sustainability attitudes.
- The constraint mechanism is mathematically elegant. Introducing per-user $λ_u$ (tolerance thresholds) and $α_u$ (penalty weights) as learnable parameters within a Lagrangian framework is principled and interpretable. The joint training objective that balances rating loss, ranking loss, sustainability penalty, and $L_2$ regularization is well-constructed and the hyperparameter ablation is reasonable.
- SusDiet is a genuinely valuable resource. Covering 149,035 recipes, 131 countries, four sustainability dimensions, and 744,138 real user interactions, it addresses a clear gap in the community. The GPT-assisted annotation pipeline with rigorous human evaluation (91–97% acceptable outputs across tasks) is a reasonable and well-documented methodology. Releasing this dataset would have lasting value independent of the modeling contribution.
- Table 3 shows consistently high $R^2$ (0.987–0.993) and low MAE/RMSE across all four sustainability prediction tasks, indicating that the MoE encoder reliably encodes diverse sustainability attributes and provides a strong foundation for the downstream constraint mechanism.

**Weakness**
- This is the most significant gap in the paper. The constraint thresholds $λ_u$ and trade-off weights $α_u$ are described as "learnable parameters" inferred from historical interaction data, but the mechanism by which they are learned is never formally described. The paper briefly acknowledges this in the limitations section but provides no empirical analysis. The key questions left is that how heterogeneous are learned $λ_u$ values across users? Do they meaningfully vary, or does the model collapse to similar values for most users? What is the distribution of constraint tightness across the user population? Without this analysis, the core claim that constraints are genuinely personalized remains largely asserted.
- In Table 5, the preference-only baseline produces worse environmental scores than the user's own historical behavior. This means the preference learning module, without constraints, actively pushes recommendations toward more unsustainable options. While the paper frames this as motivation for the constraint module, it raises a deeper question: is the preference module introducing recommendation bias that the constraint module is then correcting after the fact, rather than the two modules operating as designed complements? This interaction deserves explicit analysis.
- The abstract and introduction explicitly position cultural acceptability as one of four pillars of sustainable diets. SusDiet includes country labels across 131 nations, but no cultural acceptability indicator is constructed or used during training or evaluation. This is a meaningful inconsistency between the paper's framing and its implementation that should either be addressed or honestly scoped down in the introduction.
- The paper's practical framing like bridging the "intention-behavior gap," creating a "nudging effect," facilitating sustainable eating as "a seamless part of a user's daily routine" makes strong claims about real-world behavioral impact. None of these claims are substantiated with any user study, online experiment, or even simulated behavioral analysis.
- GRAPE is the most directly relevant baseline and achieves better NDCG@5 (0.1049 vs. 0.0120 for the proposed method). The paper does not adequately explain this large gap. At N=5, GRAPE outperforms the proposed method on ranking accuracy by nearly an order of magnitude. The proposed method recovers competitiveness at N=10 and N=20, but the discrepancy at N=5 is stark and unexplained. The paper attributes the overall results to favorable sustainability performance, but the trade-off between sustainability and recommendation precision deserves much more careful analysis than is currently provided.
- $R^2$ values of 0.987–0.993 on the sustainability prediction sub-task are unusually high and could indicate target leakage or overly similar train/test recipe distributions. The paper does not describe how train/test splits are constructed for this evaluation. Clarification is needed.

---

> ### Author Rebuttal · Authors · 2026-03-31
>
> We thank the reviewer for insightful comments. We address your concerns below.
>
> **1. Constraint mechanism**
>
> $λ$ and $α$ are learned via joint optimization. The sustainability penalty term is $α_u^\top (z_r - λ_u)$, and a ReLU operation is applied when computing the loss. Concretely, the update of $λ_u$ and $α_u$ is driven by the gradients of the penalty term with respect to observed user–recipe interactions. For users with historically higher sustainability scores, more interacted recipes satisfy $z_r > λ_u$, so $(z_r - λ_u)$ is positive more often. This more frequently pushes the model to reduce violation by increasing $λ_u$ and decreasing $α_u$.
>
> To verify personalization, we further analyze the distribution of learned user-specific constraint thresholds $λ_u$, including mean, standard deviation and percentiles (P10, P50, P90):
> ||mean|std|P10|P50|P90|
> |-------|-----|-----|-----|-----|-----|
> |λ_Nutr.|20.93|4.10|15.19|22.36|24.83|
> |λ_Envi.|67.25|47.40|46.11|63.63|92.83|
> |λ_Econ.|9.14|4.10|6.94|7.84|12.09|
> |λ_Anim.|9.37|20.80|3.80|5.10|13.65|
>
> The spread between percentiles indicates that $λ$ varies significantly across users, demonstrating that the learned constraints are personalized and capture heterogeneous tolerance levels.
>
> **2. Baseline produces worse environmental scores**
>
> The observation in Table 5 is based on a single user case study and reflects only an individual instance. When aggregated across users, the preference-only baseline achieves an average environmental score of 70.86, very close to the users' historical average of 70.38. This indicates that the preference learning module, when used alone, primarily reproduces historical behavior without shifting toward more or less sustainable outcomes. After incorporating the constraint module, the average environmental score improves to 66.29, indicating that the constraint module provides more sustainable recommendations.
>
> For the interaction between two modules, during inference, items are ranked by combining the preference score and the constraint penalty $\hat{y} _{u,r} - p _{u,r}$. During training, the preference loss and constraint loss are jointly optimized through backpropagation.
>
> **3. Cultural acceptability**
>
> Cultural acceptability is captured through learned user preferences. NDCG and Recall are considered to be indicators for the degree of user acceptability. Our model learns user preferences from historical interactions, which naturally encode users' cultural familiarity and culinary inclinations. In addition, country information is included as item-side features in the preference learning, allowing the model to account for cultural context.
>
> **4. Real‑world impact**
>
> We made the claims about practical impact as conceptual motivations based on validated results from offline experiment in the Conclusion Section, and included case studies (Table 5 and Appendix C.3) to illustrate the model's behavior. Future online or user-centered studies could validate our real-world behavioral effects, which is pointed out in this Section.
>
> **5. NDCG@5 of GRAPE**
>
> The reported NDCG@5 for GRAPE (0.1049) was a transcription error; the correct value is 0.01049. Our method achieves a higher NDCG@5 (0.0120) than GRAPE and attains the best or second-best performance on most recommendation metrics.
>
> **6. High R² in sustainability prediction**
>
> Splits are constructed via random sampling with a ratio of 80% training/10% validating/10% test. We perform deduplication to ensure no identical recipe appears in both training and test datasets. High R² is not due to target leakage, but reflects that sustainability indicators are largely determined by ingredient compositions, which can be learned accurately given sufficient training data.
>
> **7. Bias in the dataset**
>
> While the dataset is dominated by Western recipes, our method is general and learns from available recipe–sustainability associations. If more diverse data are collected, similar processing pipelines can be applied to further expand the dataset. Then, the recommendation method can be naturally extended to broader regions and better generalization across different populations.
>
> **8. Diversity of recommendations**
>
> We evaluate diversity via Item Coverage (IC) @ k, defined as the proportion of unique recipes for top-k recommendation to users relative to the entire recipe set, comparing the user interaction history, the preference-only model and our method. The results are shown below:
> |Metric|History|Preference-only|Ours|
> |-----|-------|---------------|----|
> |IC@5|0.61|0.42|0.36|
> |IC@10|0.61|0.55|0.45|
> |IC@20|0.61|0.72|0.55|
>
> Incorporating sustainability constraints leads to a moderate reduction in coverage. However, our method maintains substantial coverage that remains comparable to the scale of observed user interactions. While the constraints prioritize more sustainable recipes, they do not strictly exclude less sustainable ones. Such items are typically ranked lower but may still be recommended.

---

> > ### Author Rebuttal · Reviewer_WaLm · 2026-04-03
> >
> > I thank the authors for their thorough rebuttal, which successfully addresses my primary technical concerns and leads me to increase my scores for Soundness and Significance. Specifically, the correction of the transcription error regarding GRAPE's NDCG@5 (from 0.1049 to 0.01049) resolves the most significant discrepancy in the original results, demonstrating that the proposed method is indeed competitive with state-of-the-art baselines. Furthermore, the provided empirical distribution of learned user-specific constraint thresholds ($\lambda_u$) offers the necessary evidence that the framework captures meaningful inter-individual heterogeneity rather than collapsing to global averages. I also appreciate the inclusion of Item Coverage (IC) metrics to mitigate concerns regarding recommendation collapse and the clarification on the data split methodology. While the integration of cultural acceptability remains somewhat implicit within the preference learning module, the authors' response significantly bolsters the paper's empirical grounding and technical clarity.

---

> > > ### Author Response · Authors · 2026-04-07
> > >
> > > Thank you again for your positive acknowledgment and for confirming that all concerns have been fully addressed. We greatly appreciate your constructive review.

---

### Official Review · Reviewer_nt8b · 2026-03-12

**Soundness:** 3
**Presentation:** 3
**Significance:** 3
**Originality:** 3
**Overall Recommendation:** 4
**Confidence:** 4

**Summary:**

This paper focus on diet recommendation, aiming to recommend recipes that match user taste preferences while satisfying sustainability constraints. It formulates the task as a constraint-aware optimization problem and proposes two modules for learning sustainability score and preference, respectively. In addition, the authors construct SusDiet, a large-scale dataset with sustainability annotations and user–recipe interactions, and demonstrate improved recommendation performance and sustainability outcomes.

**Compliance With Llm Reviewing Policy:**

Affirmed.

**Key Questions For Authors:**

How can you ensure the dataset quality in SusDiet?

**Limitations:**

yes

**Strengths And Weaknesses:**

Weaknesses
1. SusDiet appears to rely heavily on LLMs in the data processing pipeline. For example, using GPT for unit normalization and quantity conversion may introduce reliability issues. Food measurements and portion sizes can vary significantly across countries and cultures (e.g., a “cup” in Japan may differ from a “cup” in the U.S.).
2. I would suggest including more LLM-based baselines, for example [1].

[1]: Mohbat, F. and Zaki, M.J., 2025, July. Kerl: Knowledge-enhanced personalized recipe recommendation using large language models. In Proceedings of the 63rd Annual Meeting of the Association for Computational Linguistics (Volume 1: Long Papers) (pp. 19125-19141).

---

> ### Author Rebuttal · Authors · 2026-03-31
>
> Thank you for your thoughtful feedback. We address your concerns below.
>
> **1. Reliability of LLM-based Processing and Dataset Quality**
>
> LLMs are only used as auxiliary tools in the data pipeline, and all key steps are supported by human-in-the-loop validation and structured processing, rather than relying on GPT outputs directly. The full data construction process is detailed in Appendix B.
>
> Specifically, each stage combines GPT assistance with human validation:
>
> - Ingredient phrases are decomposed into structured triplets (quantity, unit, ingredient) using GPT, followed by validation
> - Ingredient names are grouped by GPT and manually verified, resulting in 6,943 standardized ingredients
> - Units are consolidated from 645 distinct unit expressions into 64 unified units, with each mapping manually verified.
> - Quantity estimation is performed after structured parsing to improve consistency
> - Sustainability indicators are aligned to external databases via GPT-assisted disambiguation with human verification
> - User–recipe interactions are filtered through strict matching and metadata validation
>
> This structured design (e.g., two-stage parsing → estimation) is empirically more reliable than direct estimation (3.36 vs. 2.85 in human evaluation). For key GPT-assisted components, we further conduct targeted human evaluation (4-point scale). The results show strong reliability: 97% acceptable for phrase parsing, 91% for quantity estimation, and 94% for sustainability mapping, with average scores of 3.776, 3.664, and 3.352, respectively (Appendix B.7). These results demonstrate that the pipeline achieves high data quality beyond direct LLM outputs.
>
> Regarding cross-cultural variation (e.g., differences in “cup”), such variation is not explicitly modeled in the current version. The current pipeline normalizes ambiguous units into standardized representations to reduce variance, and human evaluation shows that this process achieves highly acceptable standardization quality. To explicitly incorporate cultural differences, one may need to introduce region-aware normalization, such as conditioning unit-to-weight conversion on geographic metadata or recipe origin, and developing culture-specific measurement mappings. Such an approach would require processing hundreds of times more entries, consuming substantial GPT generation resources, representing a promising direction for constructing a more precise and context-aware dataset.
>
> **2. More LLM-based baselines**
>
> To evaluate LLM-based baselines on our dataset, we constructed the dataset following a pipeline inspired by prior work, including
>
> - Recipes, ingredients, and user interactions were processed to form structured knowledge graphs and user behavior sequences.
> - Each recipe graph contains title, ingredients, nutrition information, and tags (derived from country).
> - From user sequences, we generated QA-style samples with constraints on sustainability attributes.
>
> We evaluated both Phi-3-mini-128k [1] and the KERL [2] model on this dataset, where the former is the baseline of KERL. Preliminary results are summarized below:
>
> | Metric | Phi-3 | KERL    | Ours    |
> | ------------------ | ------------------------ | ------- | ------- |
> | NDCG@10  | 0.0035| 0.0160  | 0.0136  |
> | Recall@10  | 0.0087| 0.0243  | 0.0223  |
> | Nutrition@10↑      | 48.5782 | 43.5833 | 39.1429 |
> | Environment@10↓    | 125.0952| 76.8827 | 67.8675 |
> | Economy@10↓        | 7.5879 | 9.5769  | 4.8684  |
> | Animal Welfare@10↓ | 4.3617  | 20.1999 | 2.9186  |
>
> As shown in the table, Phi-3 performs poorly on recommendation metrics and struggles to balance user preferences with multiple sustainability dimensions, likely due to its limited understanding of these aspects. KERL slightly outperforms our approach on recommendation metrics, but falls significantly behind on environmental, economic, and animal welfare metrics. This may be because it does not enforce strict constraints and instead allows greater trade-offs in user preference dimensions. KERL performs well on nutrition, likely due to being originally trained on datasets containing nutritional information. Note that due to time and computational resource constraints, models were used without fine-tuning. Given the relevance of KERL to our study, we will include a citation and discussion of this work in the final version.
>
>
> *[1] Marah Abdin, Sam Ade Jacobs, Ammar Ahmad Awan, Jyoti Aneja, Ahmed Awadallah, Hany Awadalla, Nguyen Bach, Amit Bahree, Arash Bakhtiari, Harkirat Behl, et al. 2024. Phi-3 technical report: A highly capable language model locally on your phone. Microsoft, MSR-TR-2024-12, Aug. 2024.*
>
> *[2] Mohbat F, Zaki M J. Kerl: Knowledge-enhanced personalized recipe recommendation using large language models. Proceedings of the 63rd Annual Meeting of the Association for Computational Linguistics (Volume 1: Long Papers). 2025: 19125-19141.*

---

> > ### Author Rebuttal · Reviewer_nt8b · 2026-04-01
> >
> > I still have some concern on benchmark quality (e.g. units conversion), so I will keep my score.

---

> > > ### Author Response · Authors · 2026-04-02
> > >
> > > We truly appreciate your feedback and understand your concern regarding potential imprecision arising from cultural differences in units and measurement standards.
> > >
> > > We want to emphasize that constructing SusDiet has already involved substantial effort and careful quality control. As detailed in Appendix B, all units were manually normalized into standardized categories, and all ingredient names were standardized through human verification. Furthermore, we conducted systematic human evaluation across all key GPT-assisted processing steps and performed a comparative analysis of one-stage versus two-stage LLM pipelines to ensure reliability.
> > >
> > > To our knowledge, SusDiet is the first large-scale dataset that integrates broad sustainability dimensions (geographic, environmental, nutritional, and economic aspects) aligned with user interactions for personalized sustainable diet recommendation. The dataset will be released publicly in the final version, and we believe it can provide valuable support for future research in sustainable development.
> > >
> > > While we have implemented rigorous standardization protocols, we acknowledge that cultural differences in units may introduce certain imprecision. However, we believe SusDiet offers a solid foundation, and future work can further refine unit standardization based on our resource.
> > >
> > > Thank you again for your constructive feedback.

---

### Official Review · Reviewer_EUxS · 2026-03-13

**Soundness:** 2
**Presentation:** 4
**Significance:** 3
**Originality:** 3
**Overall Recommendation:** 4
**Confidence:** 3

**Summary:**

This paper proposed a novel task for recipe recommendation with individual-specific sustainability factors, including nutrition adequacy, cost, cultural acceptability, and environmental impact, as constraints. A new dataset comprising of 150k recipes annotated with ingredients, regions, and sustainability metrics, were also constructed for evaluation. Results show a trade-off between NDCG and sustainability factors.

**Compliance With Llm Reviewing Policy:**

Affirmed.

**Key Questions For Authors:**

1. What are the supports that nutritional adequacy, cost, and cultural acceptability, are not already included in user preferences?
2. Why does this work require new architectures for sustainability representation and preference learning? Is using LLMs for sustainability representation and conventional reccomendation model not possible?

**Limitations:**

Yes

**Strengths And Weaknesses:**

Strengths

Presentation:
- The paper is clear and well-structured.

Significance:
- This work put individual-specific sustainability factors into focus in food recommendation task, modelling these factors as constraints. This can be further implement in other recommendation system.
- The dataset of 150k recipes and sustainable factors annotated is a valuable asset.

Originality:
- Framing individual-specific sustainability preference is novel.

Weaknesses

Soundness:
- The proposed method still sacrifices NDCG for sustainability factors.
- With the information about the recipe's sustainability factors available, conventional might be able to provide better recommendation while maintaining sustainability impacts. This scenario should also be provided in the experimental results.
- Rather than designing and training a new architecture, using LLMs to provide the information should yield more accurate recipe sustainability representation (similar to dataset construction shown in Appendix B.2).
- Comparison between proposed architecture for both recipe sustainability representation and preference learning and similar two-stage process using conventional methods would support the paper even more.

Significance:
- Most of the factors, including nutritional adequacy, cost, and cultural acceptability, are already included in user choices and might have no significance to model as constraints.

---

> ### Author Rebuttal · Authors · 2026-03-31
>
> We sincerely thank the reviewer for the constructive suggestions. We address each concern in detail below.
>
> **1. The trade-off between NDCG and sustainability**
>
> Our goal is not to strictly maximize NDCG, but to model a principled trade-off between preference satisfaction and sustainability constraints. Our framework introduces explicit constraints on nutritional adequacy, cost and environmental impact, which restrict the feasible recommendation space. Importantly, results show that our method achieves competitive NDCG compared to baselines, while consistently improving multiple sustainability dimensions. The degradation in NDCG is marginal, whereas the gains in sustainability are substantial. This suggests that our method achieves balanced and practically meaningful recommendations.
>
> **2. Incorporation of sustainability information into baselines**
>
> For baselines that support item-side attribute modeling (ICLRec, MSSR, HAFR, FGCN and GRAPE), we have incorporated sustainability indicators of recipes as input features. KNN was excluded from this setting as it does not support item-side attributes. As shown in Table 2, these baselines with the sustainability information do not outperform our method in terms of recommendation accuracy and sustainability metrics.
>
> **3. Comparing with LLM**
>
> We conducted experiments using Llama 3-8B to generate sustainability representations for all recipes. The generated representations were used as item‑side features in most baselines (ICLRec, MSSR, HAFR, FGCN, GRAPE) and also in our framework (replacing the sustainability representation stage while keeping the constraint mechanism). Results on Top‑20 recommendations are shown below.
>
> ||ICLRec(LLM)|MSSR(LLM)|HAFR(LLM)|FGCN(LLM)|GRAPE(LLM)|Ours(LLM)|Ours|
> |-------|-----------|---------|---------|---------|----------|---------|-------|
> |NDCG|0.0124|0.0154|0.0092|0.0105|0.0137|0.0148|0.0159|
> |Recall|0.0248|0.0201|0.0159|0.0224|0.0264| 0.0293|0.0316|
> |Nutr.↑|33.6432|36.0231|36.6532|32.6542|36.4352|37.1464|37.5927|
> |Envi.↓|63.3256|66.7433|64.6534|71.5634|68.6534|67.6534|66.2894|
> |Econ.↓|6.6743|7.0234|7.7643 |8.4534|6.4764|6.3462|4.9445|
> |Anim.↓| 10.4234| 9.3454|12.3244|12.2345|8.9753|5.7434|3.0255|
>
> The results show that our method consistently outperforms all LLM-based variants in recommendation accuracy and sustainability performance, indicating that LLMs alone are not sufficient for this task. ①When directly using LLMs for sustainability representation (Ours(LLM)), direct LLM outputs lack structured and reliable modeling, and cannot explicitly capture fine-grained sustainability dimensions in a single step, leading to worse performance than our method. This observation is consistent with Appendix B.7: in the data construction stage, we compared direct LLM generation with a structured two-stage approach, where the latter significantly outperformed the former (3.36 vs. 2.85 in human evaluation). This demonstrates that LLM outputs are inherently less reliable without additional learning. ②Replacing the sustainability representations in conventional baselines with LLM-generated features (baseline(LLM)) does not consistently improve performance and even degrades some metrics, further indicating that LLMs are not well‑suited for learning reliable sustainability representations alone.
>
> This task requires that representations are structured, controllable, and aligned with downstream objectives. The proposed architecture is therefore necessary to enable joint optimization and effective interaction between preference learning and sustainability modeling.
>
> **4. Whether sustainability factors are included in user preferences**
>
> Our interaction data consists of user ratings on online recipe platforms without nutritional or cost information. Therefore users cannot directly perceive nutritional adequacy or economic cost. These dimensions cannot be reliably learned from preferences alone. We therefore model these dimensions as explicit constraints to guide recommendations toward more sustainable options. For cultural acceptability, we model it as user preference in our framework.
>
> To further demonstrate the necessity of modeling each sustainability dimension as an explicit constraint, we conduct ablation studies by removing the constraint for each dimension individually while keeping the rest of the framework unchanged. The results on Top-20 recommendations are summarized below. The results show that removing any single constraint consistently degrades performance on the corresponding sustainability dimension, and in some cases also negatively affects other dimensions. In contrast, the full model achieves the best overall balance across all dimensions.
>
> ||Nutr.↑|Envi.↓|Econ.↓|Anim.↓|
> |----------|-------|-------|-------|-------|
> |w/o Nutr.|35.3454|66.1432|4.9464|3.1235|
> |w/o Envi.|37.6464|70.0443|6.5433|4.7532|
> |w/o Econ.|36.1467|67.4213|7.0745|4.9742|
> |w/o Anim.|37.4642|68.6434|6.3134|8.4523|
> |Full Model|37.5927|66.2894|4.9445|3.0255|

---

### Decision · Program_Chairs · 2026-04-30

**Decision:**

Accept (regular)

**Comment:**

This work introduces a framework for sustainable diet recommendation that treats sustainability as user-specific constraints rather than preferences. Reviewers found this formulation novel, well-motivated, and conceptually meaningful, and viewed the SusDiet dataset as a valuable contribution.

Initial concerns focused on the learning and interpretation of personalized constraints, inconsistencies in baseline comparisons, limited analysis of high-conflict users, potential recommendation collapse, and dataset quality. The rebuttal addressed most of these issues: it corrected the GRAPE baseline, demonstrated meaningful variation in learned constraints, added analysis for high-conflict users, and included diversity metrics. These clarifications resolved the majority of reviewer concerns.

Some limitations remain, particularly regarding dataset construction and the implicit treatment of cultural factors. Claims about real-world impact should therefore be interpreted with appropriate caution.

Overall, reviewer sentiment is positive. The paper makes a solid contribution in formulation, methodology, and dataset, and I recommend acceptance.